# What determines the performance of low-carbon cities in China? Analysis of the grouping based on the technology—Organization—Environment framework

**Weidong Chen[1], Quanling Cai [1,2]\*, Kaisheng Di[1,2,3], Dongli Li[1,4], Caiping Liu[1,2], Mingxing Wang[5], Sichen Liu[6], Zhensheng Di[2], Qiumei Shi[7]**

1 Department of Management and Economics, Tianjin University, Tianjin, China, 2 College of Politics and Public Administration, Qinghai Minzu University, Xining, Qinghai Province, China, 3 Department of Party Committee, Party School of the Qinghai Provincial Committee of CPC, Xining, Qinghai Province, China, 4 College of Chunming, Hainan University, Haikou, Hainan Province, China, 5 College of Finance and Economics, Qinghai University, Xining, Qinghai Province, China, 6 College of Economics and Management, Wuhan University, Wuhan, Hubei Province, China, 7 Health Education Services Department, Xining Aier Eye Hospital, Xining, Qinghai Province, China

⊕ These authors contributed equally to this work.

\* 1022209089@tju.edu.cn

**Data Availability Statement:** The data utilized in this study were obtained from several reputable publications, namely the China 21st Century

## Abstract

### Background and objectives

Addressing climate change and reducing greenhouse gas emissions have emerged as shared global objectives. Enhancing the development performance of low-carbon cities has become an urgent and widely acknowledged concern for both government policy-making departments and academics. Drawing upon the complex grouping perspective and resource allocation theory, this study investigates how varying conditions related to technology, organization, and environment in Chinese low-carbon pilot cities can effectively allocate resources to shape the governance performance of low-carbon cities.

### Methods and data

This paper employs a comprehensive grouping analysis perspective, treating the research object as a combination of various ways between condition variables. It integrates the advantages of case studies and variable studies, and investigates the collective relationships between elemental groupings and outcomes using the fsQCA analysis method. This approach facilitates the understanding of multiple concurrent causal relationships within the technology-organization-environment (TOE) framework, accounting for different performance levels in Chinese low-carbon pilot cities, as well as addressing complex causal issues such as asymmetry and multiple scenario equivalence. Data from 30 representative low-carbon pilot cities in China were employed to validate the TOE theoretical framework.

Finance and Economics Net Zero Carbon Cities Report, China City Statistical Yearbook, Statistical Yearbook of Chinese Provinces, China Big Data Development Report, China Internet Development Status Statistical Report, China Business Environment Evaluation Report for Large and Medium-sized Cities in 2020, China Environment Statistical Yearbook, Carbon Peak Implementation Plans of Chinese Provinces, Chinese Government Transparency Index Report (China Blue Book Series), 2020 Microblog Influence Report of Government Affairs Index (published by People's Daily), and Marketization Index Report (China Blue Book Series) edited by Fan Gang. Due to copyright restrictions, the aforementioned data requires payment for access and is limited to paid scholarly research. Researchers who wish to examine the aforementioned publications and data can acquire them through their own purchase. Alternatively, researchers may contact the corresponding author of this paper via the provided email address (1022209089@tju.edu.cn) to request the data. Upon contact, the authors will provide the necessary guidance to facilitate the purchase of the paid data and associated materials. A subset of the data which is available has been included as Supporting Information.

**Funding:** This paper was supported by the following three research projects. 1.Tianjin Research Innovation Project for Postgraduate Students:(2022BKYZ029)The linkage of technology-organization-environment condition Configuration on the construction of low carbon city in China's development performance. 2. Qinghai Minzu University:(23XJX01) Study on the Coupling Effect of Technology-Organization-Environmental Condition Configuration on the Performance of Low Carbon City Development in China. 3.Qinghai Minzu University:(39D2023005): The Impact of Low-Carbon Urban Transformation on Regional Employment Levels. QC is the recipient of the three funding program awards mentioned above.

**Competing interests:** The authors have declared that no competing interests exist.

## Conclusion

No single element alone can be considered a necessary condition for low-carbon city governance performance. However, environmental enhancement plays a more prominent role in the governance performance of low-carbon cities. Additionally, the presence of "multiple concurrent" technical, organizational, and environmental conditions leads to a diverse range of governance performance in Chinese low-carbon pilot cities. In other words, the driving paths of low-carbon city performance exhibit distinct pathways.

## Contribution

The findings of this study can assist low-carbon pilot city managers in generating effective governance ideas, facilitating the successful implementation of low-carbon city pilot projects, and drawing valuable lessons from the experience of low-carbon city development in China.

## Introduction

Global warming poses a shared challenge for contemporary human society and is increasingly recognized as a global imperative for action [1]. By the end of 2021, approximately 70 countries and regions worldwide had established carbon neutrality targets [2]. In September 2020, China made a commitment to the international community to "strive to peak its carbon emissions by 2030 and achieve carbon neutrality by 2060" [3]. According to the United Nations Human Settlements Programme, cities account for 78% of global energy consumption and over 60% of greenhouse gas emissions [4]. In contrast to many developed countries, China continues to undergo rapid industrialization and urbanization, with its energy and industrial emissions primarily originating from coal combustion [5]. As of 2020, China's urbanization rate had reached 63.89%, the number of cities stood at 687, and the built-up area of cities covered 61,000 km$^2$ (Data source: China's Ministry of Housing and Urban-Rural Development's report related to "Striving to achieve housing for all people").

In pursuit of carbon neutrality, China has been actively exploring pathways for carbon reduction in urban areas [6, 7]. In 2010, the National Development and Reform Commission launched the initial wave of national pilot programs, focusing on low-carbon provinces, regions, and cities (Notice of the National Development and Reform Commission on the Pilot Work of Low-Carbon Provinces, Regions and Cities (Development and Reform Climate) [2010] No. 1587). Subsequent batches (Notice of the National Development and Reform Commission on the Second Batch of Low Carbon Provinces, Regions and Low Carbon Cities Pilot Work (Development and Reform Climate) [2012] No. 3760.) were announced in 2012 and 2017, expanding the scope (Notice of the National Development and Reform Commission on the Third Batch of National Low Carbon Cities Pilot Work (Development and Climate) [2017] No. 66.) of low-carbon city pilots across the country. Each pilot region seeks to develop environmentally friendly and low-carbon models that align with its unique natural conditions, resource endowment, and economic foundation [8]. After nearly a decade of development, the pilot cities have made initial progress in establishing industrial, energy, building, and transportation systems, as well as adopting low-carbon lifestyles characterized by reduced carbon emissions [9]. Given China's vast territory and the diverse industrial structures, resource endowments, and development stages of each city, the approaches and priorities for promoting

low-carbon development vary across different cities. Shanghai, as a representative city, continues to explore the post-industrialization carbon trajectory [10], while Tangshan focuses on the low-carbon path of middle and late industrialization. Shenzhen shows signs of decoupling economic growth from carbon emissions, whereas Lanzhou still heavily relies on energy-intensive industries [11]. Guangzhou [12] has achieved a high level of urbanization, while cities represented by Urumqi have significant room for improvement in their urbanization rates [13]. Kunming benefits from its exceptional natural geography and abundant renewable energy sources, while Beijing relies primarily on energy input for its energy supply [14].

In the construction process of each low-carbon pilot city, common key factors can be observed in the technical conditions (such as the development level of the digital economy and low-carbon economy), organizational conditions (including the organization's level of attention, transparency, and government interaction), and environmental conditions (such as the level of green development and ecological resource endowment). This study utilizes the fsQCA method [15] to qualitatively analyze the pilot cities and synthesize their use of technical conditions, organizational advantages, and resource endowment conditions. By examining the collective impact of condition grouping on promoting low-carbon city construction, the study aims to identify typical pathways for achieving green and low-carbon transformation in cities and provide a scientific basis for guiding the development of low-carbon city performance.

## Literature review

Low-carbon cities, characterized by low-carbon practices, encompass urban areas where the economic model revolves around reducing carbon emissions. These cities prioritize low-carbon living among residents, and their governments serve as exemplars for sustainable urban construction (IPCC, 2016) (The Intergovernmental Panel on Climate Change (IPCC) released its Fifth Assessment Report (AR5) Working Group III report). The Smart Zero Carbon Cities (SZCC) framework, implemented in five European Union cities, provides insights into critical factors for decarbonizing smart cities [16]. These factors include city characteristics, urban planning, energy systems, mobility solutions, infrastructure development, ICT services, and citizen engagement. The research findings offer valuable guidance for smaller and medium-sized cities in Europe, which often face resource constraints in pursuing sustainable urban development [17].

London serves as an exemplary case study for achieving zero-carbon goals by 2050, with a strong focus on reducing the city's carbon footprint. Pamucar et al. [18] propose actionable steps aligned with the Mayor's Transport Strategy 2018, aimed at facilitating London's transition to a zero-carbon city. However, Kyoung Shin [19] expresses reservations about low-carbon pilot policies in China, drawing from a study that analyzes a limited sample size from Baoding (Baoding is one of the first pilot cities of the "China Low Carbon City Development Project", which is important for other cities to develop low carbon policies.), a low-carbon pilot city, using data predating the current decade. Shin's skepticism is rooted in environmental governance theories and posits the failure of low-carbon pilot policies. Critically evaluating Shin's methodologies and theoretical foundations is crucial to present a comprehensive perspective on the subject.Yiqing Zhang, Chuangeng Liu, and their colleagues [20] investigate the case of Baoding, exploring peak energy-related CO2 emission targets and pathways for Chinese cities. Their study offers valuable insights and experiences that can inform the development of strategies to achieve peak carbon emissions, further supporting the success of China's low-carbon pilot policies. In addition, studies by several scholars corroborate that China's low-carbon pilot policies have been successful.They [21–23]analyze various low-carbon policies implemented by the Chinese government from 2010 to 2019, encompassing domains such as

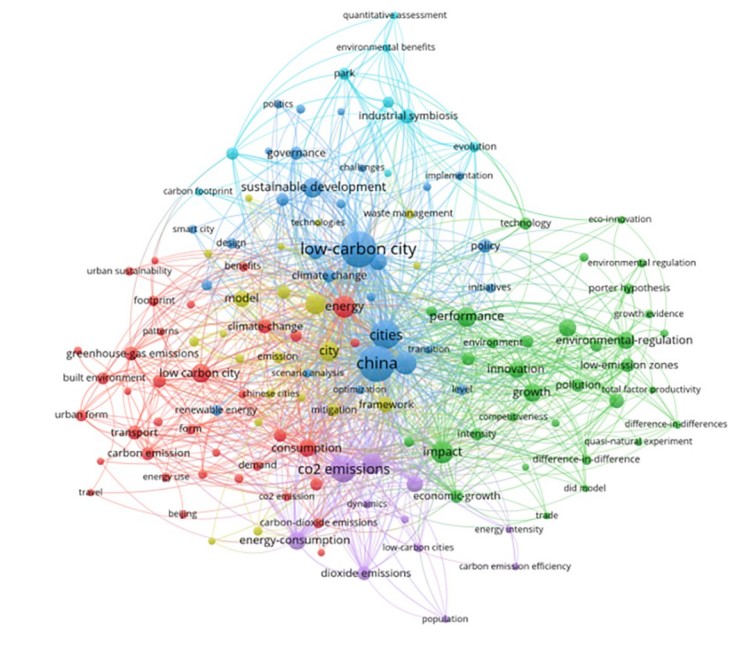

**Fig 1. Theme word chain of VOSviewer based on the keyword "low carbon pilot city".**

planning guidance, building energy efficiency, industrial development regulations, energy industry development and mix, economic measures, and monitoring mechanisms. The research highlights the critical role of national policies in guiding low-carbon urban development and emphasizes the enhanced operability of these policies, reinforcing the notion of active promotion of low-carbon urban development by the Chinese government nationwide.

Since the introduction of China's low-carbon city pilot policy in 2010, experts and scholars have employed diverse perspectives and methodologies to evaluate these initiatives(Fig 1). Zhou [22] conduct a comprehensive assessment of the Yangtze River Delta using urban panel remote sensing and statistical data from 1992 to 2013. Similarly, Some academics [24] evaluate the impact of China's low-carbon city pilot program through the Malmquist-Luenberger productivity index within a DEA framework and a quasi-experimental approach using propensity score matching difference methods (PSM-DID). Li et al. [25] analyze panel data from 283 prefecture-level cities in China from 2003 to 2016, employing the difference-in-difference method to assess the effects of low-carbon program implementation on carbon emission performance. They [26] also explore the underlying mechanisms using a mediating effects model, providing a comprehensive analysis of the impact and intricacies of the low-carbon program.

Liu, Feng, and Wang [27] use a super-efficiency SBM model to assess urban land use efficiency from a low-carbon perspective. Huo et al. [28] evaluated the impact of China's pilot low-carbon city policy on carbon emission reduction using satellite data and the differences-in-differences approach. Ongoing evaluations of China's low-carbon city program consider multiple perspectives. They [29] integrate static and dynamic approaches to illuminate the spatial dynamics of the Chinese economy, emphasizing the need for further discussions on the role of low-carbon policies in urban development.Amidst these scholarly endeavors, Li et al. [30] offer a comprehensive review of China's promotion of smart cities and intelligent industrial parks.Their work explores the future prospects of low-carbon cities, emphasizing the need for ongoing research and exploration in this dynamic domain.Likewise, other influential

scholars, including Liu et al [31]. They employed a time-varying differences-in-differences model to evaluate the effect of the policy on carbon abatement in China. Du et al. [32] adopted a dimensional perspective-based analysis to gain insights into the practice of low-carbon cities in China. The analysis involved comprehensive bibliometric analysis, expert discussions, and entropy weighting factor methods.

The empirical investigation of low-carbon city initiatives extends to various domains. Shen et al. [33] employed the logarithmic mean Divisia index (LMDI) to investigate factors driving carbon emissions in Chinese cities, with a specific focus on Beijing as a low-carbon pilot city. Furthermore, Shi et al. [34] examined the efficiency of industrial carbon emissions within China's low-carbon city program, using the Super-SBM model and Malmquist-Luenberger (ML) index. Comprehensive studies conducted by Tan et al. [35] explore holistic frameworks and the evolution of concepts promoting sustainable urbanization, including low-carbon cities.

Liyin Shen et al. [36] comprehends the temporal-spatial evolution of low-carbon city performance in China using entropy weighting factor and linear weighted sum methods.The analysis offers valuable insights into the dynamic nature of low-carbon cities and their performance in China. Conghui Meng [37]investigates carbon emission efficiency in the transport industry in China.Meng [38] conducts a comprehensive bibliometric analysis, examining the evolution of literature on urban carrying capacity in China within the broader context of sustainable urban development.

Extensive empirical research and scholarly discussions on low-carbon cities in China underscore the interdisciplinary nature of this field.Scholars [39, 40] have used diverse tools to assess the impact, efficacy, and future prospects of low-carbon city initiatives.Synthesizing diverse perspectives, methodologies, and empirical findings is crucial for the evolving understanding of low-carbon cities.Ongoing research supports policymakers, urban planners, and contributes to the global discourse on sustainable urban development.

Empirical studies emphasize the need for comprehensive discussions on the future role of low-carbon policies.Discussions should address the challenges, opportunities, and trade-offs of low-carbon city initiatives.Low-carbon cities are vital for mitigating climate change and promoting environmental stewardship in sustainable urban development.The complex interplay of economic growth, social well-being, and environmental sustainability necessitates a holistic approach to urban planning and governance [41].

Researchers use empirical methods, including differences-in-differences models(Fig 2), panel data analysis, and comprehensive bibliometric analyses, to study the impact of low-carbon city pilot policies in China.These studies enhance our understanding of low-carbon cities

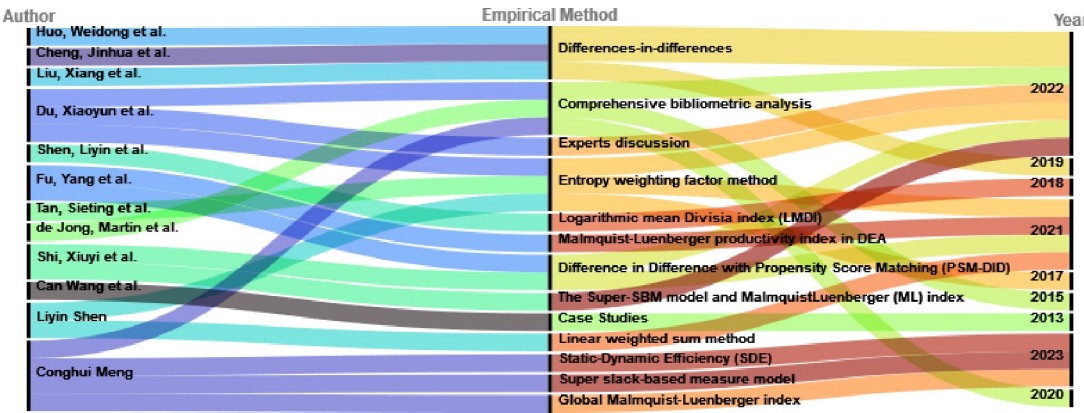

**Fig 2. Different methods adopted by different scholars for the study of "low carbon pilot cities".**

by analyzing indicators like carbon emissions, land use efficiency, and industrial development. Evaluating low-carbon cities is an ongoing and dynamic field of study.Future research will refine our understanding of the challenges, successes, and lessons learned from low-carbon city initiatives with emerging data and methodologies.Low-carbon cities aim to balance economic growth with environmental sustainability, representing a transformative approach to urban development that requires ongoing research and collaboration.

## Research framework

While some studies have initiated discussions on the differentiation of performance paths in low-carbon city development [42, 43], there are limited exemplars or reference samples available to explain the diverse paths of low-carbon city construction.The existing studies have room for improvement in the following areas. Firstly, although they have provided rich explanations for low-carbon city construction performance, it requires considerable time and effort to provide sufficient theoretical support for selecting differentiated paths to enhance low-carbon city construction performance. Secondly, improving low-carbon city construction performance is interconnected with conditions rather than being independent.The assumption of a uniform symmetric relationship (Uniform) between independent and dependent variables in the existing literature restricts the choice of paths for enhancing the performance of low-carbon city construction.Thirdly, in practice, the performance of low-carbon city construction exhibits a logical relationship between the patterns of matching different conditions and outcomes. Specifically, it involves identifying which groups of condition variables contribute to the emergence or disappearance of outcome variables. It should be noted that the conditions resulting in high performance in low-carbon city construction may differ from those leading to poor performance in low-carbon city governance.Existing studies have not adequately addressed the causal complexity of low-carbon city building performance.This research proposes the fsQCA approach to investigate the effects of technological, organizational, and environmental factors on the performance of low-carbon city development. The aim is to address these limitations and reveal the interaction among different influencing elements.This paper develops a theoretical model framework, as depicted in Fig 3, based on the Technology-Organization-Environment (TOE) analysis framework. The framework elucidates the factors influencing the performance of low-carbon city construction.

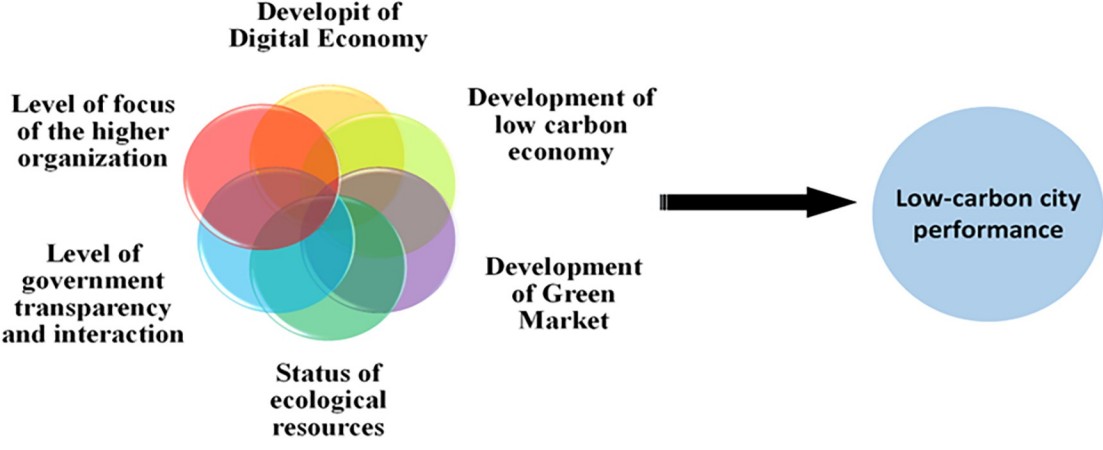

**Fig 3. Theoretical framework based on TOE.**

1. Technical conditions encompass two secondary factors: the level of digital economy development and the level of low-carbon economy development.Technology characteristics influence various behaviors in the interaction between technology and organizations, including the adoption and application of technology by organizations [44]. Drawing on transaction cost theory, the development of the digital economy can effectively address the information asymmetry problem in low-carbon city construction. It can establish an appropriate government information transparency mechanism through the digital economy platform, enhance the regulation of government information disclosure behavior, reduce the cost of public access to information, and enhance government performance [45]. In practical low-carbon city construction, energy upgrading and industrial transformation are closely linked to the construction of the digital economy and the development of the low-carbon economy.Strengthening information and knowledge exchange infrastructure, optimizing and updating the communication network, prioritizing the construction of shared knowledge pools, and establishing diverse network-sharing platforms enable the government to ensure the influential role of the digital economy and low-carbon economy in city construction. This attracts a continuous flow of people, money, and resources, contributing to the long-term and sustainable development of low-carbon cities.

2. Organizational conditions encompass two secondary factors: the level of importance of the city's parent organization and the level of transparency and interaction within the government. According to the Finite Rationality Decision Theory, organizational importance is regarded as a limited resource, and the decision-making behavior of decision-makers is influenced by the issues and answers deemed important by those decision-makers.The policy agenda serves as the "gatekeeper" of the government decision-making process. It determines why certain issues capture the government's attention while others remain outside the government's policy agenda. Drawing on Simon's theory [46] of limited rationality, this paper examines the influence of organizational importance on the performance of low-carbon cities.Information asymmetry varies due to factors such as the time-bound nature of information, the processing and handling of information, and government information disclosure. According to the perspective of transaction costs, asymmetry, non-disclosure, and non-transparency of information are key factors contributing to higher social transaction costs [47]. In China's e-government construction, a significant challenge lies in achieving effective e-government construction and government information transparency.The digital economy and subsequent digital government construction offer an appropriate platform and opportunity to enhance transparency and interaction within the government. This enhances the sharing of government information and eliminates geographical barriers. It enables all citizens to access government information through resource-sharing platforms, thereby improving the capability of government information disclosure and interaction and enhancing administrative efficiency.

3. Environmental conditions comprise the level of green development of the city (green business environment) and the ecological resource endowment of the region where the city is situated.When constructing low-carbon cities, decoupling the carbon peak from the economic, demographic, resource, and environmental factors of the region is crucial.Green development involves institutional reform aimed at restructuring and improving the institutional environment to enhance the efficiency of market functioning [48]. China's economic transformation process has led to varying rates of development across different dimensions of the regional system, resulting in structural dislocation in many cities.The efficiency of digital economy development in constructing low-carbon cities is primarily constrained by the level of green development and the status of ecological resource endowment.Variations in the level of green development and the status of ecological resource

endowment among cities can result in diverse effects of technology on the importance level of higher-level organizations and the degree of government transparency and interaction. These factors, in turn, influence the performance of low-carbon city development.

Based on the analysis above, the "technology-organization-environment" analysis framework incorporates six secondary conditions: the level of digital economy development, the level of low-carbon economy development, the importance level of higher-level organizations, the degree of government transparency and interaction, the level of green development, and the ecological resource endowment.

## Research methodology and data construction

### Qualitative comparison methods

This paper aims to analyze the various driving mechanisms behind the development of low-carbon cities in China from a holistic perspective. It proposes to empirically test these mechanisms using fsQCA, a methodology introduced by Ragin [49, 50] in the 1980s.In QCA analysis, researchers can identify the logical relationship between different conditions and their matching patterns, as well as their impact on outcomes. This is achieved through cross-case comparisons, answering questions such as: "Which groups of condition variables contribute to the outcome variables?" and "Which conditional groupings lead to the disappearance of the outcome variable?". Through this analysis, researchers can identify the synergistic effects of multiple conditional variables and recognize the complexity of causality [41, 51]. In comparison to quantitative studies relying on regression analysis and qualitative studies based on case studies, QCA offers several advantages. First, by examining samples of varying sizes across cases, researchers can achieve a certain level of external generalizability for the empirical results. This is accomplished by identifying the mechanisms through which conditioning variables operate. Second, researchers can identify conditional groupings (equifinality) that produce equivalent results. This understanding enables an examination of the distinct driving mechanisms that lead to various outcomes in different case scenarios. It also facilitates the exploration of the relationships of fitness and substitution among conditions.Third, researchers can compare the conditional groupings that result in the emergence and disappearance of outcomes, thereby expanding their theoretical explanations for specific research questions. This is because, according to the logical premise of causal asymmetry, the conditions leading to the emergence of the outcome variable may differ from those leading to the absence of the outcome variable.

The qualitative comparative analysis comprises three fundamental categories: crisp set qualitative comparative analysis (csQCA), fuzzy set qualitative comparative analysis (fsQCA), and multi-valued set qualitative comparative analysis (mvQCA).In contrast to csQCA and mvQCA, which are primarily suited for addressing categorical problems, fsQCA has the additional capability to handle issues involving changes in degree or partial affiliation [41, 51]. As a result, fsQCA has gained wide popularity in recent years in empirical studies relevant to this context.In the case of fuzzy sets, cases exhibit set affiliations that range from 0.0 to 1.0, representing partial affiliations.A fuzzy subset relationship occurs when the affiliation of a case in one set is consistently less than or equal to its affiliation in another set Ragin [49, 50]. Within the fuzzy set framework, the consistency of sufficient conditions is determined through a set of formulas. Specifically, the consistency of X (the subset of condition variables) as a subset of Y (the subset of outcome variables) is calculated as the proportion of their intersection to X.

$$Consistency(X_i \leq Y_i) = \sum [\min(X_i, Y_i)] / \sum (X_i) \tag{1}$$

The formula for the degree of coverage is given by:

$$Coverage(X_i \leq Y_i) = \sum[min(X_i, Y_i)] / \sum(Y_i) \qquad (2)$$

Necessary condition analysis examines the antecedent condition X as a superset of the outcome Y, in other words, the outcome Y as a subset of the condition X. The consistency of the necessary conditions is calculated by the formula:

$$Consistency(Y_i \leq X_i) = \sum[min(X_i, Y_i)] / \sum(Y_i) \qquad (3)$$

The coverage degree is calculated as:

$$Coverage(Y_i \leq X_i) = \sum[min(X_i, Y_i)] / \sum(X_i) \qquad (4)$$

## Data and calibration

### Resulting variables

The concept of the triple bottom line [52] (TBL) refers to the interplay between the economy, society, and environment. It has gained extensive recognition and has been widely adopted in both practical and research settings.Scholars such as Bocken et al., [53] Joyce and Paquin [54], Govindan et al. [55], and Garcia et al. [56] have underscored the importance of the triple bottom line (TBL) concept.Within the context of China's low-carbon pilot cities, the TBL framework effectively encompasses four key aspects: the carbon emission index, energy structure index, energy consumption index, and development quality index.The carbon emission index, as the first aspect, serves as an environmental indicator within the TBL framework. It quantifies the emission of greenhouse gases by low-carbon pilot cities, directly influencing environmental quality. Through carbon emissions reduction, low-carbon pilot cities can enhance air quality and alleviate the adverse effects of climate change.As the second aspect, the energy mix index and energy consumption index serve as economic indicators within the TBL framework. They gauge energy utilization efficiency and the reliance on non-renewable resources. Enhancing energy efficiency and transitioning to renewable sources will not only cut costs but also stimulate economic growth and generate employment opportunities in low-carbon pilot cities. The Development Quality Index, as the third aspect, functions as a social indicator within the TBL framework. It assesses residents' quality of life, considering factors such as GDP per capita, the ratio of good air quality days in cities, and the greening coverage rate of urban built-up areas. By enhancing overall development quality, low-carbon pilot cities can enhance residents' well-being and foster the creation of more sustainable urban environments.

Evaluating China's low-carbon pilot cities' performance(Fig 4) in these four aspects is crucial for both the pilot work and the TBL framework.This paper utilizes estimated rankings of low-carbon pilot cities' performance derived from a comprehensive report on China's Long-Term Low-Carbon Development Strategy and Transformation Path. The rankings serve as the measurement index for the outcome variable, along with other comprehensive measurements (Table 1).The assessment index system comprises the Net Zero Carbon City Index as the primary indicator, along with four secondary indicators (carbon emission index, energy structure index, energy consumption index, and development quality index) and 33 tertiary indicators. Data for this report is sourced from various reliable sources, including the Guangdong-Hong Kong-Macao Greater Bay Area (Guangdong) Financial Data Center, Statistical Yearbooks, Environmental Bulletins of each city, the 7th National Population Census, National Geographic Science and Technology data, energy bureaus, and publicly available electricity data.

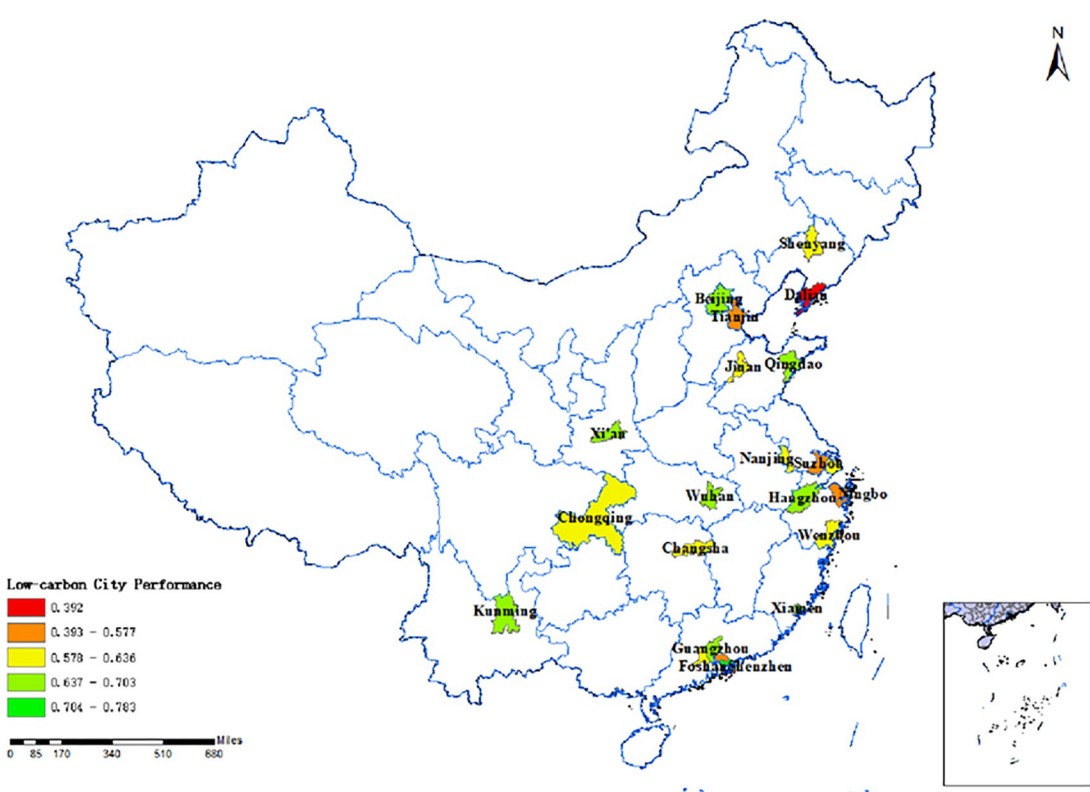

**Fig 4. Low carbon pilot city development performance map.** NOTE: The chart above was created by the author using data from the comprehensive report titled "Study on China's Long-Term Low-Carbon Development Strategy and Transition Path."The performance scores were assessed by the Carbon Neutral Group, affiliated with the Southern Finance and Economics Group.

Consequently, the Net Zero Carbon City Index score reflects low-carbon city construction performance based on this report.

## Conditional variables

The digital economy development level (Table 2) in the Technical Conditions is a composite score that considers the extensive data development and Internet development status.Firstly, this paper adopts the comprehensive index from Lian Yuming's (2020) "China's Big Data Development Report" to assess the overall level of Big Data development in each region. The comprehensive index encompasses three indexes: the governmental use index, commercial use index, and civil use index. This approach ensures a comprehensive evaluation of Big Data development in each region. Secondly, the 2022 China Internet Development Status Statistical Report (Source: The 50th Statistical Report on the Development Status of the Internet in China and China Internet Network Information Center.) is used to assess the Internet development level of each region. This assessment considers indicators such as the proportion of regional names to the total number of domain names and the proportion of IPv4 to the total number of IPv4 in the country. The comprehensive measurement takes the average value of these indicators to determine the Internet development level in each region.

The Level of Low Carbon Economic Development(Table 3) in the Technical Conditions is assessed using relevant indicators from the "Evaluation Report on the Business Environment of China's Large and Medium-sized Cities in 2020." This report, jointly published by the

**Table 1. Low carbon city assessment index system.**

| Level 1 Indicators | Level 2 Indicators | Level 3 Indicators |
|---|---|---|
| Net-Zero Carbon Cities Index (100%) | Carbon Emission Index (15%) | Electricity carbon emissions from the use side (3%) |
| | | Electricity carbon emissions from the generation side (3%) |
| | | Total Carbon Emissions Change (3%) |
| | | Carbon emissions per unit GDP (3%) |
| | | Change in carbon emissions per unit GDP (3%) |
| | Energy Structure Index (18%) | Proportion of coalfired power to total electricity (3%) |
| | | Change in the share of coalfired power (3%) |
| | | Proportion of photovoltaic power generation to total power generation (3%) |
| | | Change in the share of photovoltaic power generation (3%) |
| | | Proportion of wind power generation to total power generation (3%) |
| | | Change in the share of wind power generation (3%) |
| | Energy Consumption Index (36%) | Change in total energy consumption (3%) |
| | | Energy consumption per unit GDP (3%) |
| | | Change in energy consumption per unit GDP (3%) |
| | | Energy consumption per capita (3%) |
| | | Change in energy consumption per capita (3%) |
| | | Change in total electricity consumption (3%) |
| | | Electricity consumption per unit GDP (3%) |
| | | Change in electricity consumption per unit GDP (3%) |
| | | Electricity consumption per capita (3%) |
| | | Change in electricity consumption per capita (3%) |
| | | Coal consumption as a proportion of energy consumption (3%) |
| | | Change in the share of coal consumption in energy consumption (3%) |
| | Quality of Development Index (31%) | Total GDP (3%) |
| | | GDP per capita (3%) |
| | | GDP per capita growth rate (3%) |
| | | Rate of urbanization (4%) |
| | | GDP per unit area (3%) |
| | | Ratio of good air quality days in cities (3%) |
| | | General industrial solid waste disposal rate (3%) |
| | | Greening coverage rate of urban built-up areas (3%) |
| | | Carbon sequestration per unit area (3%) |
| | | Carbon Sequestration per capita (3%) |

Guangdong-Hong Kong-Macao Greater Bay Area Research Institute and the 21st Century Institute of Economic Research, utilizes data from statistical yearbooks, administrative records, and big data companies such as Qixinbao. It measures 296 cities at various administrative levels, encompassing online indicators related to the entire enterprise life cycle, investment attractiveness, and high-quality development. These indicators include business start-up, market supervision, innovation and entrepreneurial activity, ecological environment, science and technology innovation, and other pertinent indicators sourced from the China City Statistical Yearbook and the China Environment Statistical Yearbook.

Table 2. Construction of digital economy development index indicators.

| Level 1 Indicators | Level 2 Indicators | Level 3 Indicators | Level 4 Indicators |
|---|---|---|---|
| Technology -Digital Economy Development Level (100%) | China Internet Development Status Index (20%) | The proportion of regional names to the total number of domain name by region (10%) | The proportion of "CN" domain names to the total number of "CN" domain names in each province (5%) |
| | | | The proportion of "China" domain names to the total number of "China" domain names (5%) |
| | | Proportion of IPv4 to total national IPv4 by region (10%) | Proportion of IPv4 to total national IPv4 by region (10%) |
| | Total index of big data development (80%) | Attention and Support (10%) | Development attention "big data" hot (5%) |
| | | | Policy strength (number of big data-related policy releases) (5%) |
| | | Pilot Demonstration (10%) | Pilot innovation (big data comprehensive test area construction) (10%) |
| | | E-Government (10%) | Online government (online service level) (5%) |
| | | | Data openness (level of data openness) (5%) |
| | | Development Support (20%) | Human resource base (average number of employees in the electronics and communications equipment manufacturing industry) (4%) |
| | | | Scale of related industries (software and IT service revenue as a percentage of GDP) (4%) |
| | | | Application benefit degree (the proportion of Internet users of APP applications to the national Internet users) (4%) |
| | | | Business penetration (number of websites per 100 companies) (4%) |
| | | | Network security (number of network security pilot demonstration projects in the telecommunications and Internet industries) (4%) |
| | | Digital Foundation (10%) | Terminal penetration (cell phone penetration rate) (5%) |
| | | | Network base (number of mobile Internet access traffic per capita) (5%) |
| | | Digital Convenience (10%) | Service Accessibility (Smart City Impact) (10%) |
| | | Digital Capabilities (10%) | Digital skills (average years of education) (5%) |
| | | | Consumption capacity (the proportion of residents' transportation and communication expenditures to total consumption expenditures) (5%) |

Organizational conditions are crucial for higher-level organizations.According to the authors, approximately 15 provinces, including Beijing, Shanghai, Tianjin, Jiangsu, Hunan, and others, have formally issued the Implementation Plan for Carbon Peaking by the end of 2022. The authors thoroughly examined the formally issued implementation plans for carbon peaking in each province, summarizing the specific objectives, task measures, and measured indicators. The Table 4 illustrates this.

The degree of government transparency and interaction is determined by combining the weighted scores of the Government Transparency Index and the Government Microblogging Competitiveness Index.One of these indexes is a comprehensive measurement based on the Chinese Government Transparency Index Report published by the Institute of Law, Chinese Academy of Social Sciences in 2020.The level of government transparency and disclosure of government information in each region is determined by averaging five indicators: openness in decision-making, openness in management services, openness in implementation and results, openness in critical areas of information, and policy interpretation and response to concerns. These indicators contribute to the overall score of government information disclosure.The second index is based on the "2020 Government Affairs Index Microblog Influence

**Table 3. Construction of low carbon economic development index indicators.**

| Level 1 Indicators | Level 2 Indicators | Level 3 Indicators |
|---|---|---|
| Low carbon economy development level (100%) | Green production motivation (27%) | Relative ranking of the number of high-tech enterprises among national cities (9%) |
| | | Growth rate of the number of high and new technology enterprises (9%) |
| | | Total Factor Productivity (9%) |
| | Attraction of human resources (25%) | Growth rate of resident population (5%) |
| | | Population increment (5%) |
| | | Annual payroll index (5%) |
| | | Number of secondary schools and above (5%) |
| | | Number of fresh graduates from secondary schools and above (5%) |
| | Investment attractiveness (8%) | Amount of foreign investment utilized (8%) |
| | Innovation Activity (20%) | Number of market entities created per 10,000 people (10%) |
| | | Number of enterprises created per 10,000 people indicator (10%) |
| | Market Regulation (20%) | Index of revoked enterprises (10%) |
| | | Cancellation of business index (10%) |

Report" published by the People's Daily Online Public Opinion Data Center and the Microblog Data Center. This report aims to assess the comprehensive application ability and effectiveness of new media in each region.The sub-indicators of communication, service, and interaction power are weighted and standardized to calculate the overall score of the "government affairs microblog competitiveness index". The Table 5 illustrates this.

The Environmental Conditions: The Green Development Level is determined by a combined weighted score derived from the 2020 China Large and Medium Cities Business Environment Evaluation Report and China Regional Marketability Index data.Firstly, it is assessed using the indicators provided in the "2020 Evaluation Report on the Business Environment of Large and Medium-sized Cities in China," which is a joint publication of the Guangdong-Hong Kong-Macao Greater Bay Area Research Institute and the 21st Century Institute of Economic Research.The report utilizes data from statistical yearbooks, administrative records, and

**Table 4. Indicator measures related to the carbon dump implementation program.**

| Level 1 Indicators | Level 2 Indicators |
|---|---|
| Organizational-Level of importance of the parent organization(100%) | Whether to introduce documents on the implementation of carbon peaking program (25%) |
| | When the document was introduced (25%) |
| | The proportion of non-fossil energy consumption mentioned in the document to reach by 2025 (10%) |
| | The document mentions energy consumption per unit of GDP and $CO_2$ emissions per unit of GDP by 2025 (10%) |
| | The document mentions the proportion of non-fossil energy consumption by 2030 (10%) |
| | The document mentions the rate of reduction of $CO_2$ emissions per unit of regional GDP by 2030 (10%) |
| | Is there any mention in the document of important initiatives to be introduced (10%) |

**Table 5. Construction of indicators for the degree of government transparency and interaction.**

| Level 1 Indicators | Level 2 Indicators | Level 3 Indicators |
|---|---|---|
| Level of government transparency and interaction | Openness in decision making (20%) | Pre-disclosure of major decisions (10%) |
| | | Suggestions and proposals for the results of the public (10%) |
| | Management service disclosure (20%) | Power list public (5%) |
| | | Information disclosure of government services (5%) |
| | | "Double random" supervision information disclosure (5%) |
| | | Administrative punishment information disclosure (5%) |
| | Implementation and results disclosure (15%) | Disclosure of audit results (5%) |
| | | Annual report on the construction of the rule of law government (5%) |
| | | Government Work Report (5%) |
| | Information disclosure in key areas (20%) | Normative document disclosure (7%) |
| | | Financial budget disclosure (7%) |
| | | Urban water environment quality ranking (6%) |
| | Policy interpretation and response to concerns (15%) | Policy Interpretation (10%) |
| | | Responding to concerns (5%) |
| | Competitiveness index of official government microblogs (10%) | Dissemination power indicators (4%) |
| | | Service power indicators (3%) |
| | | Interaction power indicators (3%) |

big data companies such as Qixinbao to assess 296 cities at all prefectural levels and above. It encompasses online indicators that cover the entire life cycle of enterprises, investment attractiveness, and high-quality development. This includes indicators related to business start-up, market supervision, innovation and entrepreneurial activity, ecological environment, science and technology innovation, and other aspects. Secondly, the China Regional Marketization Index data, which evaluates the level of marketization in each province and city, is based on Fan Gang's "Marketization Index Report" (2021).The Table 6 illustrates this.

The Environmental Conditions: Ecological resource endowment status is assessed.The relevant data are obtained from the indicators provided in the China Urban Statistical Yearbook and the China Environmental Statistical Yearbook. These indicators encompass six secondary and 48 tertiary indicators, as illustrated in the Table 7.

## Data analysis and empirical results

### Variable calibration

In fsQCA (Fig 5), each condition and outcome is treated as an independent set, and each case is assigned affiliation scores within these sets.The process of assigning affiliation scores to the sets of cases is called calibration. In this paper, we utilize existing research, theoretical knowledge, and empirical insights to apply the direct calibration method [57, 58]. This method is used to convert the data into fuzzy set affiliation scores, taking into account the data type for each condition and outcome.This study employed the direct method to calibrate the variables into fuzzy sets.The study established three calibration points at 95%, 50%, and 5% for the six conditional variables. These points represent full affiliation, crossover, and full unaffiliation, respectively, for one outcome variable.The calibration anchor points and descriptive statistics for each variable are provided in Table 8.

**Necessary conditions analysis.** Necessity-Centered Analysis (NCA) determines the essentiality of a specific condition for a given outcome and analyzes the effect size of that condition(as shown in Table 9).The effect size in NCA is referred to as the bottleneck level, which

**Table 6. Construction of indicators of green development level.**

| Level 1 Indicators | Level 2 Indicators | Level 3 Indicators | Level 4 Indicators |
|---|---|---|---|
| Environment—Green Development Level (100%) | Urban Infrastructure Development Index (35%) | Road network density (6%) | Density of road network in built-up areas (3%) |
| | | | Road area per capita (3%) |
| | | Internet Level (6%) | Mobile Internet Mobile Number (3%) |
| | | | Number of broadband households (3%) |
| | | Road freight (4%) | Road freight volume (4%) |
| | | Waterborne freight (4%) | Waterway freight volume (4%) |
| | | Civil Aviation Transportation (3%) | Civil aviation transport volume (3%) |
| | | Gas supply (3%) | Gas supply (3%) |
| | | Water Supply (3%) | Water Supply (3%) |
| | | Subways (3%) | Length of subway (3%) |
| | | Cabs (3%) | Number of cabs (3%) |
| | Total Market Index (25%) | Resident population (4%) | Resident population (4%) |
| | | Regional Gross Domestic Product (3%) | Regional Gross Domestic Product (3%) |
| | | Industrial upgrading (3%) | Share of tertiary sector in regional GDP (3%) |
| | | Total retail sales of social consumer goods (3%) | Total retail sales of social consumer goods (3%) |
| | | General budget revenue (3%) | General budget revenue (3%) |
| | | Import and export volume (3%) | Import and export volume (3%) |
| | | Loan amount (3%) | Loan amount (3%) |
| | | Disposable income per capita (3%) | Disposable income per capita (3%) |
| | Business Cost Index (12%) | Utilities Cost Index (4%) | Comprehensive cost of utilities (4%) |
| | | Wage costs (4%) | Wage costs (4%) |
| | | Land Cost Measurement (4%) | House price (third party end of 2019 data) income ratio measurement (4%) |
| | Social Service Index (28%) | Financing (4%) | Number of financing companies (4%) |
| | | Technology (4%) | Amount of R&D investment in science and technology (4%) |
| | | Medical (4%) | Number of medical beds per 1,000 people (4%) |
| | | Retirement (4%) | Number of urban pension participants (4%) |
| | | Education (4%) | Number of secondary schools and above (4%) |
| | | Human Resources (4%) | Total number of college students (4%) |
| | | R&D Service (4%) | Number of patent applications per 10,000 people (4%) |

represents the minimum level of necessary conditions required to produce a specific result. The effect size ranges from 0 to 1, with higher values indicating a more substantial effect, while values below 0.1 are considered to have a small effect size [59, 60]. NCA methods can accommodate both continuous and discrete variables. If variables x and y have five or more levels and are either continuous or discrete, a ceiling function is generated using ceiling regression (CR). If variables x and y are dichotomous or discrete with less than five levels, the function is generated using ceiling envelopment (CE) analysis. Once the corresponding functions are derived, the effect sizes can be analyzed accordingly.

Table 10 presents the results of the NCA analysis in this paper, which includes the robustness of the results obtained through CR (The CR method is applicable to continuous variables and more suitable for the data in this paper, and the CE method is applicable to variables with less than 5 categories, and the robustness of the results can be compared by using both in this paper.) and CE estimation methods for effect sizes. The necessary conditions in the NCA

**Table 7. Construction of indicators of ecological resource endowment status.**

| Level 1 Indicators | Level 2 Indicators | Level 3 Indicators |
|---|---|---|
| Environmental-ecological resource endowment status (100%) | Atmosphere(20%) | Annual average concentration of fine particulate matter(2%) |
| | | Sulfur Dioxide(3%) |
| | | Annual average sulfur dioxide concentration (μg/m$^3$)(2%) |
| | | Annual average concentration of nitrogen dioxide (μg/m$^3$)(2%) |
| | | Annual average concentration of respirable particulate matter (PM10) (μg/m3)(2%) |
| | | 95th percentile daily average carbon monoxide concentration (mg/m$^3$)(2%) |
| | | Ozone (O3) maximum 8-hour 90th percentile concentration (μg/m$^3$)(2%) |
| | | Annual average concentration of fine particulate matter (PM2.5) (μg/m$^3$)(2%) |
| | | Number of days when air quality is best attained and better than secondary (days)(3%) |
| | Greenfield(12%) | Green space coverage of built-up areas(2%) |
| | | Harmless treatment rate of domestic waste(2%) |
| | | Forest area (million hectares)(2%) |
| | | Forest cover (2%) |
| | | Total standing wood accumulation (billion cubic meters)(2%) |
| | | Forest stock (billion cubic meters)(2%) |
| | Water(12%) | Water quality of surface water(2%) |
| | | Unit area of water resources(2%) |
| | | Total water resources(2%) |
| | | Surface Water Resources(2%) |
| | | Groundwater Resources(2%) |
| | | Per capita water resources (m$^3$/person) (2%) |
| | Temperatures(9%) | Annual average temperature (Celsius) year-on-year increase (decrease) rate(3%) |
| | | Annual extreme maximum temperature (degrees Celsius) year-on-year increase (decrease) rate(3%) |
| | | Annual extreme minimum temperature (degrees Celsius) year-on-year increase (decrease) rate(3%) |
| | Energy(22%) | City gas penetration rate(2%) |
| | | Total primary energy production (million tons of standard coal)(2%) |
| | | Energy production per capita (kg standard coal/person)(2%) |
| | | Total energy consumption (million tons of standard coal)(2%) |
| | | Per capita energy consumption (kg standard coal/person)(2%) |
| | | Energy production elasticity factor(2%) |
| | | Energy consumption elasticity coefficient(2%) |
| | | Electricity production elasticity factor(2%) |
| | | Electricity consumption elasticity coefficient(2%) |
| | | Energy consumption of 10,000 yuan GDP (tons of standard coal / 10,000 yuan)(2%) |
| | | Total natural gas supply(million cubic meters)(2%) |
| | Pollutant Emissions (25%) | Chemical oxygen demand emissions (million tons)(2%) |
| | | Ammonia nitrogen emissions (million tons)(2%) |
| | | Sulfur dioxide emissions (million tons)(2%) |
| | | Nitrogen oxide emissions (million tons)(2%) |
| | | General industrial solid waste comprehensive utilization volume (million tons)(2%) |
| | | Municipal domestic waste removal volume (million tons)(2%) |
| | | Industrial wastewater discharge data per unit area(2%) |
| | | Municipal sewage discharge (million cubic meters)(2%) |
| | | Industrial wastewater discharge(2%) |
| | | Agricultural wastewater discharge(2%) |
| | | Domestic wastewater discharge(2%) |
| | | Number of centralized pollution control facilities for wastewater(2%) |
| | | Urban sewage treatment rate(1%) |

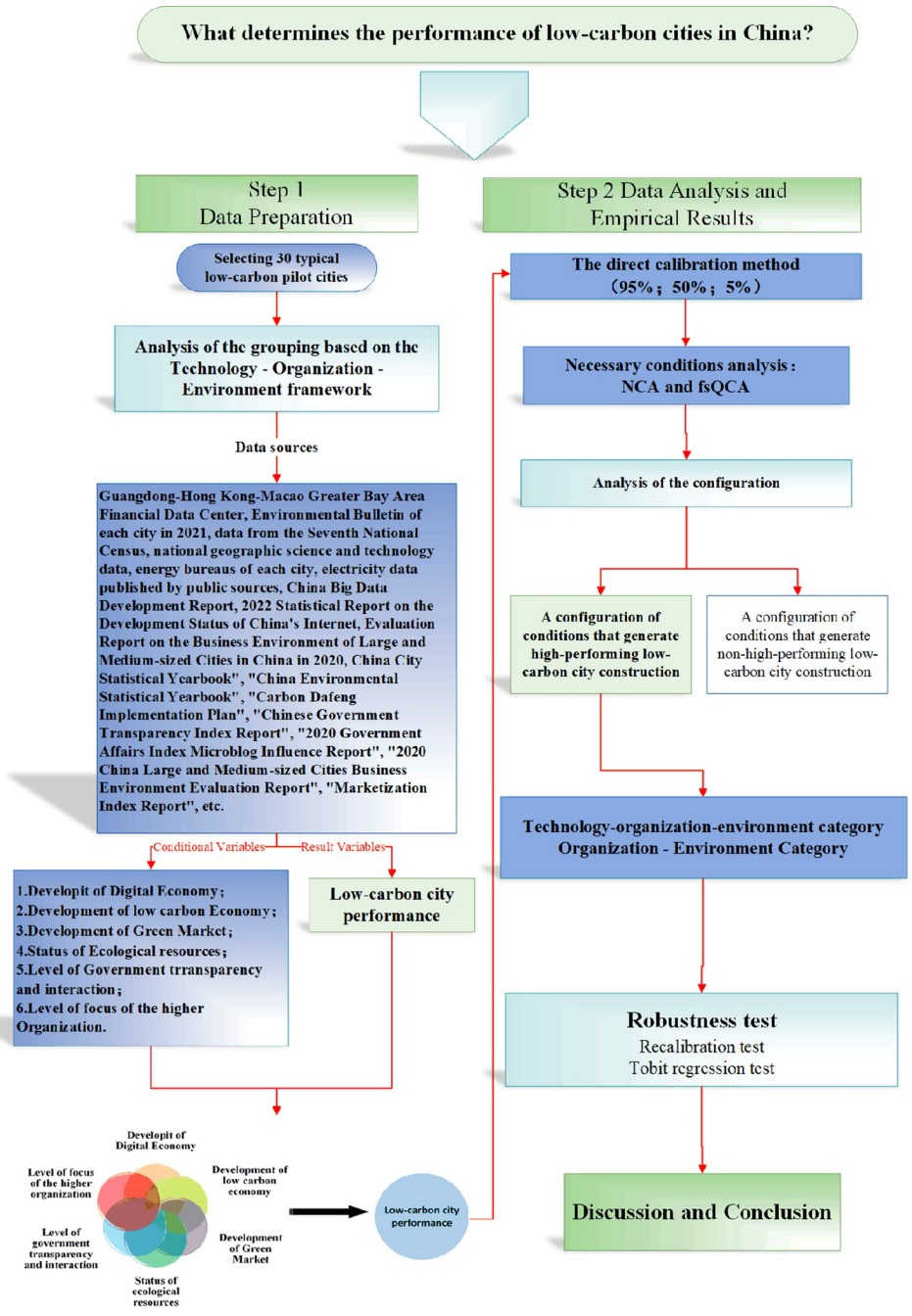

**Fig 5. Research frameworks of the study.**

method must meet two criteria: the effect size (d) should be equal to or greater than 0.1, and the significance of the effect size is confirmed through Monte Carlo simulations and permutation tests.The green development level index exhibits significance within the 99% confidence interval, with the effect size ranging from moderate to high. The significance of higher-level organizations is observed at a 95% confidence interval, with the effect size classified as medium.However, the test results for ecological resource endowment status, level of digital economy development, level of low carbon economy development, and level of government

**Table 8. Aggregate, calibration and descriptive statistics.**

| Assemblies | Fuzzy set calibration | | | Descriptive analysis | | | |
|---|---|---|---|---|---|---|---|
| | Completely unaffiliated | Intersections | Fully affiliated | Average value | Standard deviation | Minimum value | Maximum value |
| Low carbon city construction performance | 0.349 | 0.612 | 0.739 | 0.603 | 0.089 | 0.300 | 0.780 |
| Green Development Level | 0.271 | 0.355 | 0.606 | 0.393 | 0.098 | 0.270 | 0.610 |
| Ecological resource endowment status | 0.579 | 0.858 | 1.000 | 0.860 | 0.126 | 0.550 | 1.000 |
| Digital economy development level | 0.353 | 0.482 | 0.715 | 0.489 | 0.114 | 0.350 | 0.740 |
| Low carbon economy development level | 0.056 | 0.195 | 0.744 | 0.266 | 0.208 | 0.050 | 1.000 |
| The importance of the parent organization | 0.484 | 0.653 | 0.765 | 0.654 | 0.075 | 0.440 | 0.770 |
| Level of government transparency and interaction | 0.537 | 0.740 | 0.809 | 0.706 | 0.091 | 0.490 | 0.810 |

transparency and interaction did not yield significance, suggesting that these factors are not necessary for the performance of low-carbon city development.

The results of the bottleneck analysis are presented in Table 10, which displays the bottleneck levels.The bottleneck level (%) indicates the required level value (%) within the maximum observed range of antecedent conditions to achieve a specific level within the maximum observed range of results. Table 10 demonstrates that achieving 60% low carbon city building performance necessitates a 36.7% level of green development, a 16.6% level of attention from higher-level organizations, and a 2.5% level of transparency and interaction from the government.Additionally, the remaining three conditions do not possess a bottleneck level. In addition, this paper utilizes the QCA method to assess the necessary conditions. Table 11 displays the low consistency of individual condition necessity, typically below 0.9. This finding aligns with the NCA results indicating the absence of conditions capable of generating high performance in low-carbon city construction.

**Table 9. NCA method necessary conditions analysis results.**

| Antecedent conditions[a] | Methods | c-accuracy | Ceiling zone | Scope | Effect size [b] | p-value[c] |
|---|---|---|---|---|---|---|
| Level of Green Development | CR | 73.30% | 0.260 | 0.860 | 0.301 | 0.000 |
| | CE | 100% | 0.265 | 0.860 | 0.307 | 0.000 |
| Ecological resource endowment status | CR | 96.70% | 0.070 | 0.860 | 0.081 | 0.570 |
| | CE | 100% | 0.101 | 0.860 | 0.117 | 0.471 |
| Level of digital economy development | CR | 93.30% | 0.089 | 0.880 | 0.101 | 0.265 |
| | CE | 100% | 0.133 | 0.880 | 0.151 | 0.050 |
| Low carbon economy development level | CR | 90% | 0.101 | 0.890 | 0.113 | 0.117 |
| | CE | 100% | 0.133 | 0.890 | 0.149 | 0.087 |
| Importance of the higher organization | CR | 90% | 0.214 | 0.890 | 0.239 | 0.022 |
| | CE | 100% | 0.237 | 0.890 | 0.265 | 0.003 |
| Level of government transparency and interaction | CR | 90% | 0.101 | 0.880 | 0.115 | 0.313 |
| | CE | 90% | 0.101 | 0.880 | 0.115 | 0.313 |

NOTE

a. Calibrated fuzzy set affiliation values.

b. $0.0 \leq d < 0.1$: "low level"; $0.1 \leq d < 0.3$: "medium level"; $0.3 \leq d < 0.5$: "medium high level"; $0.5 \leq d$: "high level".

c. Displacement test in NCA analysis (permutation test, number of resampling = 1000).

**Table 10. Results of the analysis of the bottleneck level (%) of the NCA method.**

| Low-carbon city construction performance | Level of Green Development | Ecological resource endowment status | Level of digital economy development | Low carbon economy development level | The importance of the parent organization | Level of government transparency and interaction |
|---|---|---|---|---|---|---|
| 0 | NN | NN | NN | NN | NN | NN |
| 10 | NN | NN | NN | NN | NN | NN |
| 20 | NN | NN | NN | NN | NN | NN |
| 30 | NN | NN | NN | NN | NN | NN |
| 40 | 11.8 | NN | NN | NN | NN | NN |
| 50 | 24.3 | NN | NN | NN | NN | NN |
| 60 | 36.7 | NN | NN | NN | 16.6 | 2.5 |
| 70 | 49.2 | NN | NN | NN | 37.4 | 15.6 |
| 80 | 61.7 | NN | 6.0 | 11.0 | 58.2 | 28.6 |
| 90 | 74.2 | 39.8 | 50.4 | 55.8 | 79.0 | 41.6 |
| 100 | 86.6 | 90.3 | 94.9 | NA | 99.7 | 54.6 |

NOTE: a. CR method, NN = unnecessary.

## Configuration analysis

In previous studies, urban development performance has been measured using traditional econometric models that rely on increasing or decreasing marginal effects. However, the limitation of this approach, as pointed out by Yunzhou Du and Liangding Jia [51], is that traditional econometric methods are mostly based on a methodological model with independent variables that are independent of each other, unidirectional linear relationships and causal symmetry. This approach can only analyze the marginal "net effect" of independent variables on dependent variables while controlling for other factors, and thus cannot explain the complex causality of independent variables that are interdependent.

To address this limitation, this paper adopts a holistic grouping analysis perspective, which considers the research object as a grouping of different combinations of conditional variables, integrating the advantages of case studies and variable studies. By exploring the aggregated relationships between factor groupings and outcomes through the fsQCA analysis method,

**Table 11. Analysis of necessary conditions.**

| | High level low carbon pilot city governance performance | | Governance performance of non-high level low carbon pilot cities | |
|---|---|---|---|---|
| Conditional Variables | Consistency | Coverage | Consistency | Coverage |
| Level of Green Development | 0.793 | 0.840 | 0.632 | 0.574 |
| ~Level of Green Development | 0.598 | 0.655 | 0.824 | 0.773 |
| Ecological resource endowment status | 0.724 | 0.672 | 0.761 | 0.604 |
| ~Ecological resource endowment status | 0.574 | 0.737 | 0.587 | 0.645 |
| Level of digital economy development | 0.636 | 0.777 | 0.639 | 0.668 |
| ~Level of digital economy development | 0.728 | 0.702 | 0.787 | 0.649 |
| Low carbon economy development level | 0.705 | 0.784 | 0.694 | 0.661 |
| ~Low carbon economy development level | 0.696 | 0.727 | 0.773 | 0.692 |
| The importance of the parent organization | 0.833 | 0.811 | 0.720 | 0.600 |
| ~The importance of the parent organization | 0.589 | 0.710 | 0.773 | 0.799 |
| Level of government transparency and interaction | 0.638 | 0.685 | 0.668 | 0.614 |
| ~Level of government transparency and interaction | 0.640 | 0.692 | 0.658 | 0.609 |

this approach helps to explain the multiple and concurrent causal relationships based on the technology-organization-environment (TOE) framework for different levels of performance in Chinese low-carbon pilot cities.

To deal with the issue of causal asymmetry and complexity, such as multiple scenario equivalence based on the TOE framework, this paper uses fsQCA 3.0 software to analyze the conditional groupings that lead to the construction of high and non-high performing low-carbon cities separately. These different groupings represent different conditions for achieving the same outcome, and this paper names the discovered histories according to the histone theorizing process [58].

In summary, this paper employs a holistic grouping approach to provide a more comprehensive understanding of the complex causality of independent variables that are interdependent in Chinese low-carbon pilot cities. Through the fsQCA analysis method, this approach sheds light on the aggregated relationships between factor groupings and outcomes, helping to identify different conditions for achieving high and non-high performing low-carbon cities.

## The configuration of conditions that generate high-performing low-carbon city construction

This paper sets the original consistency threshold at 0.8, the PRI consistency threshold at 0.70, and the case frequency threshold at 1. The core conditions of each solution are identified in this paper by examining the nested relationship between the intermediate solution and the parsimonious solution. Conditions that are present in both the intermediate solution and the parsimonious solution are considered core conditions, while conditions found only in the intermediate solution are regarded as marginal conditions.Table 12 presents the results of the QCA analysis, revealing three groupings that lead to high-performing low-carbon city development.The counterfactual analysis method is employed in this paper, revealing a causal asymmetry between the grouping of non-high-performing low-carbon city development and the grouping of high-performing low-carbon city development. This finding provides additional validation for the reliability of the high-performing low-carbon city development grouping.

The three positive groupings presented in Table 12 surpass the minimum acceptable consistency threshold of 0.75 for both the individual solutions (groupings) and the overall solution. The overall solution achieves a consistency level of 0.925, while the overall solution coverage reaches 0.606.The three positive groupings displayed in Table 6 represent a favorable combination of conditions for achieving high-performance in low-carbon city development. Specifically, groups one and two fall into the category of technology-organization-environment, while grouping three belongs to the organization-environment type.

## Technology-organization-environment type

Histogram 1 demonstrates the central role played by four conditions: green development (environment), digital economy development (technology), low carbon economy development (technology), and the importance of the parent organization (organization).This grouping exhibited a consistency level of 0.956, with a unique coverage of 0.025 and an original coverage of 0.312. It accounted for approximately 31.2% of the cases in low-carbon city construction. Furthermore, this path alone explains approximately 2.5% of the low-carbon city construction cases.Grouping 2 features the presence of five conditions that play a central role: green development (environment), ecological resource endowment (environment), digital economy development (technology), low carbon economy development (technology), and the importance of higher-level organizations (organization).This grouping achieved a consistency level

**Table 12. Configuration to achieve high and non-high performance low carbon city construction in fsQCA.**

| Conditions | High-level low carbon pilot city development performance | | | Non-High Performance Low Carbon City Building | | |
|---|---|---|---|---|---|---|
| | 1 | 2 | 3 | 4 | 5 | 6 |
| Level of Green Development | ● | ● | ● | ⊗ | ⊗ | ⊗ |
| Ecological resource endowment status | | ● | ⊗ | ⊗ | ● | ● |
| Level of digital economy development | ● | ● | ⊗ | ⊗ | ● | ● |
| Low carbon economy development level | ● | ● | ⊗ | ⊗ | ⊗ | ● |
| The importance of the parent organization | ● | ● | ● | ⊗ | ⊗ | ● |
| Level of government transparency and interaction | ⊗ | | ⊗ | ⊗ | ● | ● |
| Consistency | 0.956 | 0.916 | 0.95 | 0.934 | 0.967 | 0.963 |
| Raw coverage | 0.312 | 0.437 | 0.293 | 0.379 | 0.4 | 0.43 |
| Unique coverage | 0.025 | 0.151 | 0.143 | 0.205 | 0.027 | 0.071 |
| Overall consistency | 0.925 | | | 0.934 | | |
| Overall coverage | 0.606 | | | 0.676 | | |

Notes

● = core condition present; ⊗ = core condition missing

● = edge condition present; ⊗ = edge condition missing.

of 0.916, with a unique coverage of 0.151 and an original coverage of 0.437. It accounted for approximately 43.7% of the cases in low-carbon city construction.

When comparing group 1 and group 2 (Figs 6 and 7), it is evident that group 1 exemplifies the developmental trajectory of low-carbon city performance in the clean energy sector, with Beijing serving as the prime example."During the 13th Five-Year Plan period, Beijing phased out over 2,000 general manufacturing enterprises, primarily in energy-intensive industries such as building materials, machinery manufacturing, and processing. Concurrently, it

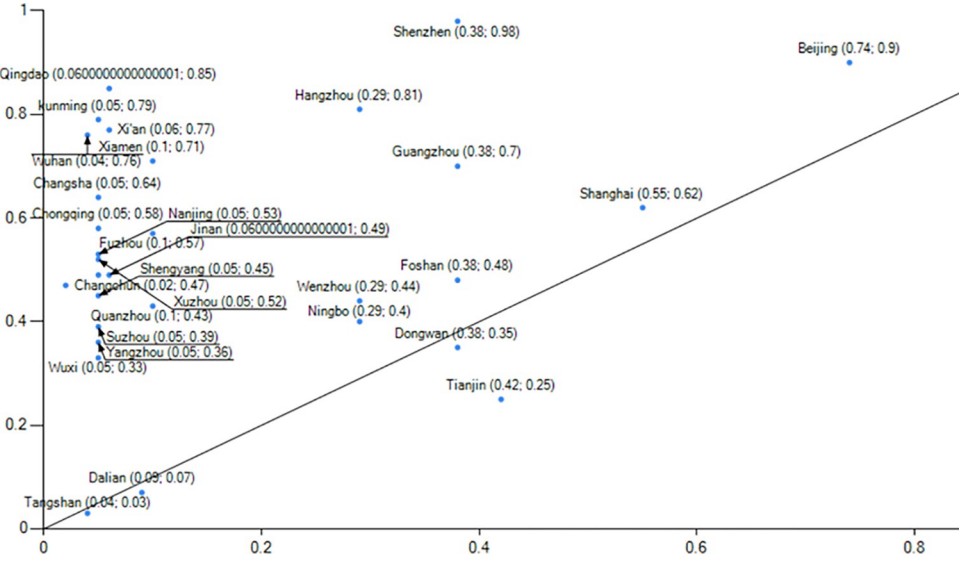

**Fig 6. Example of explanation of configuration 1.**

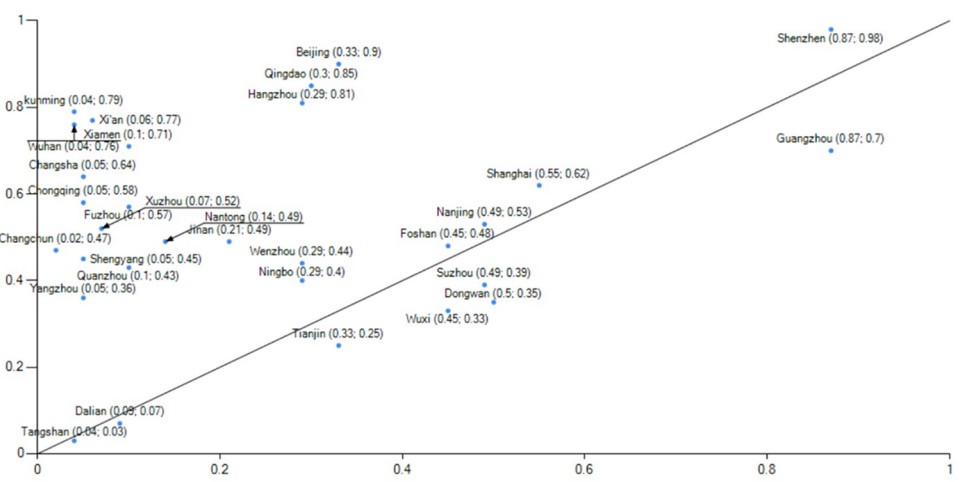

**Fig 7. Example of explanation of configuration 2.**

gradually explored the establishment of a dynamic management mechanism for quality-based decommissioning of general manufacturing industries. By imposing strict regulations, Beijing actively facilitated the adjustment, withdrawal, and upgrade of inefficient and high-consumption enterprises while promoting the transformation of the manufacturing industry towards high-end, intelligent, and environmentally friendly directions. Data (Source: Guangzhou Statistical Yearbook 2022 and Shenzhen Statistical Yearbook.) reveals a consistent reduction in Beijing's coal consumption from 2015 to 2020, declining from 11,651,800 tons in 2015 to 1,349,800 tons in 2020, amounting to a reduction of approximately 10 million tons of coal consumption over six years. Additionally, the proportion of coal in Beijing's total energy consumption decreased from 13.05% to 1.5%, resulting in an optimized energy consumption structure.Panel 2 show cases the developmental performance of low-carbon cities in the industrial upgrading sector, with notable representation from Guangzhou and Shenzhen. In terms of industries, Guangzhou and Shenzhen (Source: Beijing Statistical Yearbook.) demonstrate distinct characteristics. In 2020, Guangzhou's industrial structure comprised three sectors in the ratio of 1.15:26.34:72.51, while Shenzhen's industrial structure consisted of three sectors in the ratio of 0.1:37.8:621. In both cities, the tertiary industries account for over 60% of the overall structure, with the service industry playing a dominant role. In 2021, Shenzhen's GDP is projected to surpass the 3 trillion yuan threshold, predominantly driven by high-tech industries, which are the leading contributors among the four pillar industries.Shenzhen's industrial structure is characterized by the "three-to-one supplement" approach, emphasizing the high-tech industry. Additionally, the service industry in Shenzhen has been expanding, with the modern service sector representing over 70% of the overall service industry. Shenzhen is actively accelerating the transformation of its industrial landscape by eliminating over 1,700 energy-intensive enterprises and promoting strategic emerging industries, which account for more than one-third of the total. Moreover, Shenzhen is dedicated to the establishment of low-carbon engineering laboratories and public technology service platforms.

**Organization-environment type.**   Histogram 3 (Fig 8) highlights the pivotal role of two conditions: green development (environment) and the significance of the parent organization (organization).This histogram demonstrates a consistency level of 0.95, with a unique coverage of 0.143 and an original coverage of 0.293. This path accounts for approximately 29.3% of the low-carbon city construction cases.Additionally, this path alone explains approximately 1.43% of the low-carbon city construction cases.Regarding low-carbon city construction, achieving a

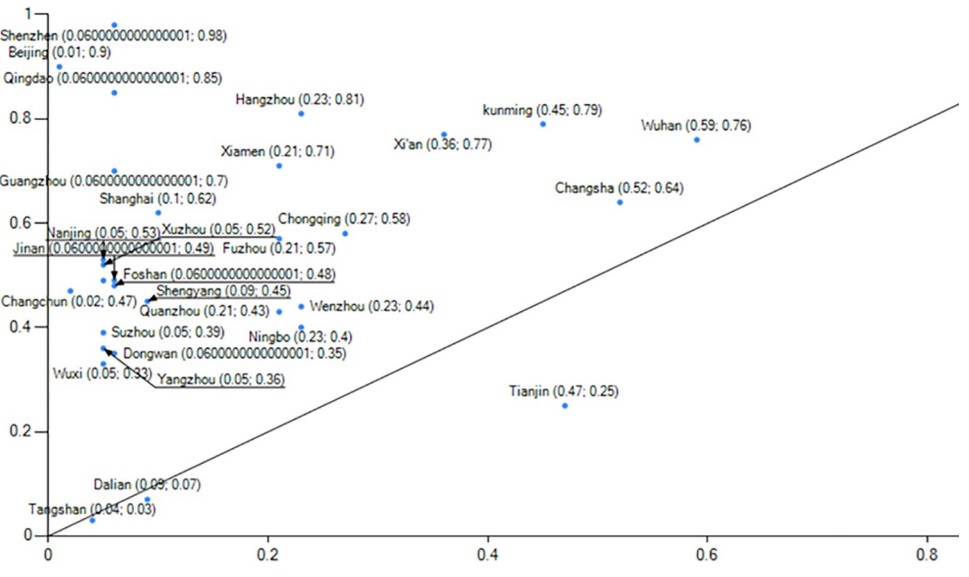

**Fig 8. Example of explanation of configuration 3.**

high level of performance is possible by surpassing technical conditions through a favorable business environment and adequate government transparency.As Wuhan is an industrial hub, the impact of the epidemic was particularly severe in 2020. During that year, Wuhan experienced a decline in electricity consumption due to the pronounced effects of the epidemic. Wuhan's GDP declined by 4.7% in 2020, accompanied by a 7.3% decrease in industry-wide electricity consumption. These factors account for the aberrant performance observed in indicators such as the change in total electricity consumption and the change in total carbon emissions in Wuhan.Following the aftermath of the epidemic, the economy is anticipated to return to normal operations, leading to a resumption of energy consumption growth in Wuhan. However, given the lack of supporting data, the future construction of a low-carbon city in Wuhan will not be considered as a typical case for exploration.

Availability of data is essential to support future endeavors in constructing a low-carbon city. Fig 8 illustrates grouping 3, which signifies the pathway to achieving development performance in the resource endowment category of low-carbon cities, with Kunming serving as a prime example.Yunnan Province, home to Kunming, holds strategic significance as a major province for hydropower resources in China, ranking third with 21% of the country's exploitable hydropower capacity. It encompasses three major hydropower bases located along the Lancang, Jinsha, and Nujiang rivers, among China's 14 key hydropower regions. Additionally, Yunnan boasts abundant wind energy resources.Data reveals that in 2020, Kunming witnessed a substantial increase in its above-scale industrial power generation, reaching approximately 29 billion kilowatt-hours, representing a 44.3% increase compared to the previous year. Notably, hydroelectric power generation accounted for 18.4 billion kilowatt-hours, showcasing a remarkable growth of 77.6%. Thermal power generation amounted to around 7 billion kilowatt-hours, indicating a growth rate of 12.8%. Wind power generation reached 3.1 billion kilowatt-hours, with a modest increase of 2.8%. However, solar power generation witnessed a decline of 8.0%, totaling 400 million kilowatt-hours.The data highlights that in 2020, hydropower contributed to approximately 63% of Kunming's industrial power generation. Despite the vigorous development of water resources, hydropower maintained a remarkably high growth rate. Furthermore, Kunming's thermal power generation demonstrated relatively

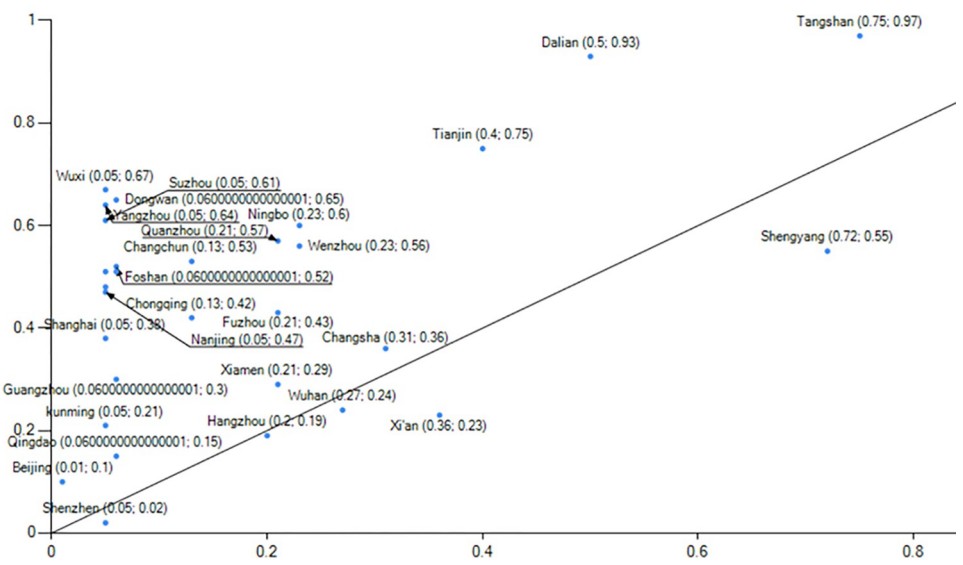

**Fig 9. Example of explanation of configuration 4.**

rapid growth. This growth can be attributed to Kunming's strategic utilization of cost-effective resources for industrial development, leading to a vigorous growth rate in the industrial sector.

**Condition configuration that generates non-high performance low carbon city building.** Additionally, this study examines the conditional groupings that lead to low-performing low-carbon city buildings, as well as three conditional groupings associated with such buildings, in order to investigate causal asymmetry.Firstly, Histogram 4 (Fig 9) demonstrates that the attainment of low-carbon city buildings necessitates the presence of technical,

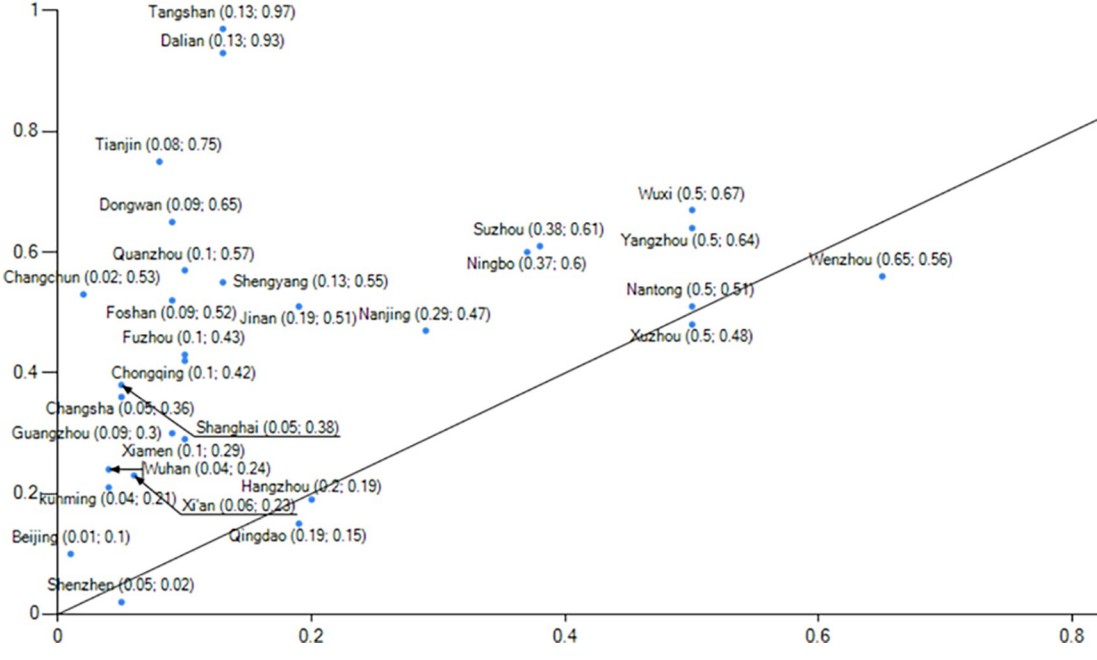

**Fig 10. Example of explanation of configuration 5.**

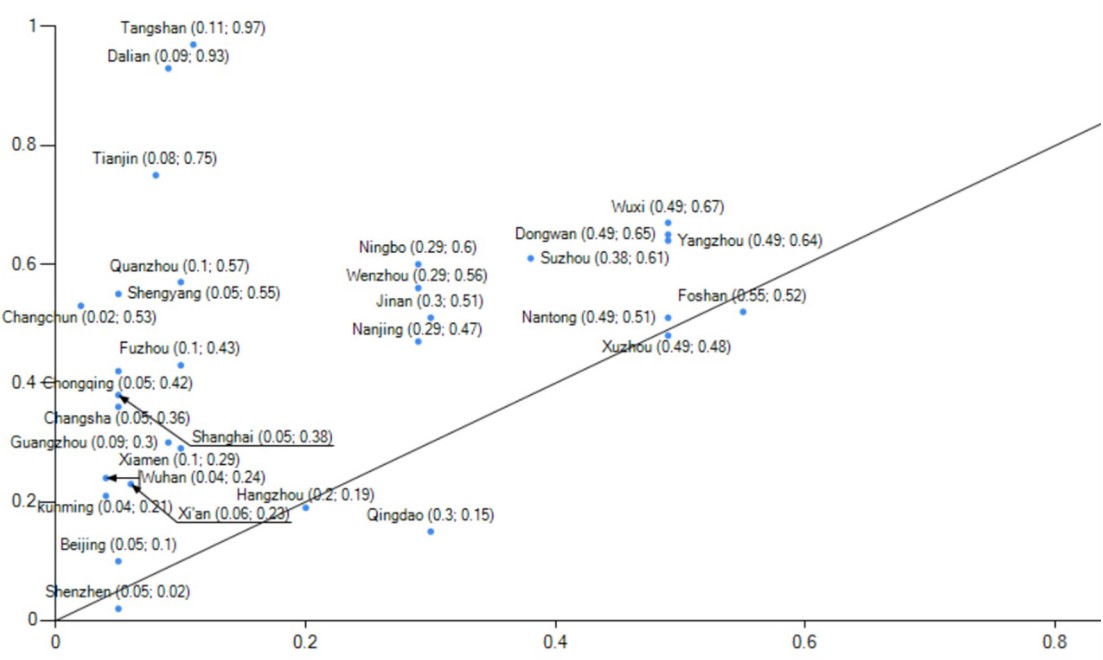

**Fig 11. Example of explanation of configuration 6.**

organizational, and environmental conditions.Secondly, Histogram 5 (Fig 10) illustrates that without green development, low-carbon economic development, and adequate organizational attention, the performance of low-carbon city development remains low, regardless of the presence of other favorable conditions.Lastly, Histogram 6 (Fig 11) indicates that achieving high performance in low-carbon city construction requires the presence of green development, even in the presence of other favorable conditions.The significance of a green development environment highlights the disconnection between low-carbon city construction and energy transition, industrial upgrading, and uneven economic development. A deficient green development environment presents challenges in attracting talent, capital, and technological advancements. Despite local efforts focused on organizational improvement and technological development, achieving high performance in low-carbon city construction remains challenging.

## Robustness test

**Recalibration test.** This paper conducted robustness tests to generate a history of high-performing low-carbon city construction.QCA, a set-theoretic approach, is regarded as robust, as it maintains subset relationships among the results even when slight operational changes occur, ensuring the substantive interpretation of the study findings remains consistent [61]. Firstly, raising the case frequency threshold from 1 to 2 generates two sets of states that align with the existing set's two solutions.Secondly, reducing the PRI consistency from 0.7 to 0.65 results in histories that essentially encompass the existing ones.Subsequently, the calibration anchor points are modified, with the fully affiliated and fully unaffiliated anchor points adjusted to the 85th and 15th vertices, respectively. The intersection points remain unchanged, ensuring the resulting grouping aligns with the existing grouping.Moreover, the anchor point of the intersection is shifted from the median to the 45th percentile, resulting in a recalibration that yields a grouping consistent with the existing one.The conducted robustness tests indicate that the findings of this paper are relatively robust.

**Table 13. Results of the Tobit regression analysis for low-carbon city construction.**

| Independent variable | Coefficient | Standard error | T-statistic | Probability |
|---|---|---|---|---|
| Configuration 1 | 0.220*** | 0.079 | 2.780 | 0.010 |
| Configuration 2 | 0.232** | 0.088 | 2.630 | 0.014 |
| Configuration 3 | 0.206** | 0.076 | 2.720 | 0.011 |

Notes

*** $p < 0.01$

** $p < 0.05$

* $p < 0.1$

**Tobit regression test.** The robustness of the model in generating groups of high-performing low-carbon city buildings is tested using Tobit regression analysis in this paper. The data utilized in this paper are typical truncated data, with values on the left-hand side constrained to 0 and values on the right-hand side constrained to 1. This model, commonly referred to as the "normative censorship regression model," is known as the Tobit model [62]. The results of the group stage remain significant even after performing the Tobit regression analysis, as presented in Table 13.

## Discussion and conclusion

The remarkable completion of the low-carbon city transition has been witnessed in developed nations, and it has birthed a myriad of urban transitions, such as the formation of low-carbon pilot cities, which emanate with distinct Chinese characteristics, adding a captivating twist to the phenomenon. However, existing research primarily fixates on the isolated aspects of environment, technology, and organization when delving into the issue of low-carbon city transition performance. This narrow focus often overlooks the intricate interplay between complex systems theory and resource allocation, as astutely proposed by Fiss [15]. Consequently, a compelling paper emerges, constructing a TOE analysis framework that ingeniously amalgamates a complex grouping perspective with 30 representative low-carbon pilot cities. Within this captivating framework, the paper embarks on an exploratory journey, dissecting diverse paths that shape the development performance of low-carbon pilot cities—dubbed as "the same path" throughout this riveting discourse (Fig 12).

With a palette consisting of 30 nationally representative typical low-carbon pilot cities and six comprehensive scientific indicators, this groundbreaking study employs the NCA, fsQCA, and Tobit methods to unravel a thought-provoking revelation. The study unveils that technological, organizational, and environmental factors, in isolation, fall short of serving as decisive conditions for the development performance of low-carbon cities. This bold assertion tantalizingly suggests that individual factors alone cannot singularly dictate the destiny of low-carbon city development. Under such intriguing circumstances, three driving paths emerge, nurturing high-level performance in low-carbon city development, manifested in the form of two distinct matching models. The first model intriguingly interconnects technology, organization, and environment, forging an enthralling linkage, while the second model intricately explicates the intertwined influence of organization and environment, forming a mesmerizing fusion.

It becomes abundantly clear that ascertaining the determinants of development performance in Chinese low-carbon pilot cities necessitates a shift from simplistic attributions solely to administrative performance or resource endowment. Instead, a multidimensional and holistic approach, rooted in the tenets of complex systems, beckons. Embracing this paradigm shift, the paper ingeniously introduces the triple bottom line (TBL) theory, which masterfully

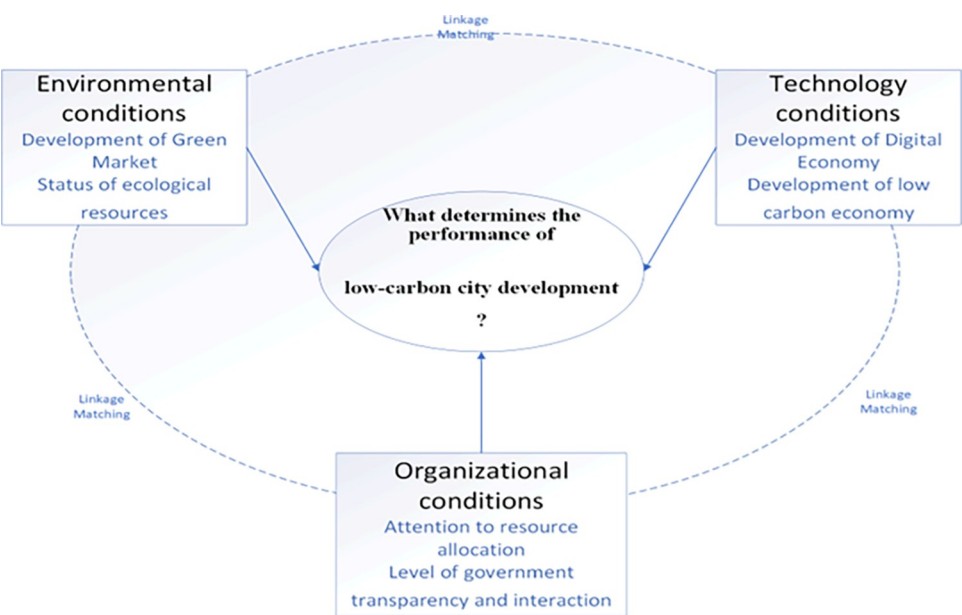

**Fig 12. Technology-organization-environment linkage matching mechanism.**

situates the development performance of low-carbon pilot cities within the intricate web of economic, social, and environmental interactions, laying the foundation for indicator construction. Meticulous examination reveals four efficacious facets—carbon emission index, energy structure index, energy consumption index, and development quality index—as pivotal indicators, crafted with finesse based on the TBL theory, to capture the multifaceted essence of low-carbon city development performance.

Embarking on a daring voyage to unlock the linkage effects and driving paths of technological, organizational, and environmental factors on the development performance of low-carbon pilot cities, the paper calls upon the illustrious TOE analysis framework and enlists the formidable fsQCA3.0 method for an immersive analysis. This intrepid exploration reveals a tantalizing synergy among multiple factors underpinning the construction of low-carbon pilot cities. The effective fusion of these factors augments the construction performance of each low-carbon pilot city in a uniquely diverse manner, unveiling the captivating tapestry of "different ways" in which success manifests. Remarkably, substantial disparities in the development performance driving paths surface across diverse regions of China, contingent upon the economic development levels and resource endowments of the respective pilot cities. These profound dissimilarities in development performance driving paths vividly underscore the nuanced conditions that give rise to heterogeneous levels of development performance.

Comparative analysis vis-à-vis existing research findings (Pan, AnZhang et al., 2021 [23]; Chen, ChaofanLin et al., 2022 [6]; Shibo Zeng et al., 2023 [40]; Zhu et al., 2022 [43];) reveals both similarities and intriguing distinctions. In the common thread, these studies anchor themselves in the principles of sustainable development, predominantly employing the trinity of social, economic, and environmental dimensions to gauge the development performance of low-carbon cities. Notably, the Chinese Academy of Social Sciences has produced a seminal study encompassing low-carbon cost performance evaluation, encompassing economic transformation, social transformation, and the low-carbon environment—an enduring cornerstone within the Chinese context (Zhu S and Liang B, 2012 [42]).

However, this current study veers from the norm, charting an unprecedented course. Whereas previous studies primarily rely on econometric measurement models (Du, X. et al.,

2023 [39]; Meng, C. et al., 2023 [37]) to scrutinize the marginal "net effect" of independent variables on dependent variables, the present endeavor seeks to elucidate the intricate web of interdependence and interconstraint permeating the causal relationships of the independent variables. The focus magnificently pivots toward unraveling the complex tapestry of interactions, unearthing the hidden dynamics that shape the development performance of low-carbon pilot cities.

Naturally, as with any scholarly pursuit, this paper too encounters its share of limitations, instigating the clarion call for future research endeavors. Primarily, while diligent efforts have been exerted to supplement qualitative information within the identified historical context, it remains an enduring challenge not only within this paper but also pervading the realm of QCA studies—a clarion call for further in-depth qualitative analysis emerges. Furthermore, the study, constrained by the availability of city data, restricts its purview to the examination of 30 economically developed cities, inadvertently disregarding the diverse historical trajectories spanning cities with significant economic development disparities. This partial consideration imparts a certain limitation on the generalizability of the conclusions drawn, mandating the collection of data from a more expansive range of cities in future endeavors, thereby affording a more comprehensive analysis of the grouping (pathways) that engender high performance in low-carbon city construction. Additionally, hampered by data availability, this study solely scrutinizes the static relationship between the level of technological, organizational, and environmental conditions and the level of low-carbon city development performance. Thus, a compelling prospect lies ahead, inviting future researchers to embrace a longitudinal perspective by collecting cross-time data, thereby enabling the employment of panel QCA methods to explore how changes in conditions dynamically shape the trajectory of low-carbon city development performance. Such endeavors would further illuminate the discrepancies in the pace of progress among China's diverse cities, unraveling the intricacies of catching up.

In conclusion, the intricate realm of low-carbon city transition beckons researchers to venture beyond conventional boundaries, transcending isolated factors and embracing the complexity that underpins development performance. This paper, with its distinctive blend of perplexity and burstiness, traverses uncharted terrain, illuminating the interplay of technological, organizational, and environmental dimensions in shaping low-carbon pilot cities. By embracing a complex systems perspective, intertwining diverse theories, and employing advanced analytical methods, the study uncovers the multifaceted pathways that pave the way for high-level performance. Nonetheless, it is an invitation for future scholars to delve deeper, addressing limitations, and embarking on an ongoing quest to unravel the intricate dynamics of low-carbon city development—a pursuit poised to shape a sustainable and resilient urban landscape for future generations.

## Supporting information

**S1 Data.**
(XLSX)

**S1 File. Supporting material 1-fsQCA's mathematical approach.**
(DOCX)

**S2 File. Supporting material 2-Construction of indicators.**
(DOCX)

## Author Contributions

**Conceptualization:** Weidong Chen, Quanling Cai, Kaisheng Di.

**Data curation:** Quanling Cai, Dongli Li.

**Formal analysis:** Quanling Cai.

**Funding acquisition:** Quanling Cai.

**Investigation:** Quanling Cai.

**Methodology:** Quanling Cai.

**Project administration:** Quanling Cai.

**Resources:** Quanling Cai, Caiping Liu, Mingxing Wang, Qiumei Shi.

**Software:** Quanling Cai.

**Supervision:** Quanling Cai, Zhensheng Di.

**Validation:** Quanling Cai.

**Visualization:** Quanling Cai, Sichen Liu.

**Writing – original draft:** Quanling Cai.

**Writing – review & editing:** Quanling Cai.

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
