## [Decision Letter · Decision Letter 0]

5 May 2023

PONE-D-23-08832What determines the performance of low-carbon cities in China?--Analysis of the grouping based on the TOE frameworkPLOS ONE

Dear Dr. Cai,

Thank you for submitting your manuscript to PLOS ONE. After careful consideration, we feel that it has merit but does not fully meet PLOS ONE’s publication criteria as it currently stands. Therefore, we invite you to submit a revised version of the manuscript that addresses the points raised during the review process.

We look forward to receiving your revised manuscript.

Kind regards,

Muhammad Farhan Bashir

Academic Editor

PLOS ONE

Journal Requirements:

3. We note that you have stated that you will provide repository information for your data at acceptance. Should your manuscript be accepted for publication, we will hold it until you provide the relevant accession numbers or DOIs necessary to access your data. If you wish to make changes to your Data Availability statement, please describe these changes in your cover letter and we will update your Data Availability statement to reflect the information you provide

Reviewers' comments:

Reviewer's Responses to Questions

**Comments to the Author**

1. Is the manuscript technically sound, and do the data support the conclusions?

Reviewer #1: Yes

Reviewer #2: Partly

Reviewer #3: No

2. Has the statistical analysis been performed appropriately and rigorously? 

Reviewer #1: Yes

Reviewer #2: Yes

Reviewer #3: No

3. Have the authors made all data underlying the findings in their manuscript fully available?

Reviewer #1: Yes

Reviewer #2: Yes

Reviewer #3: Yes

4. Is the manuscript presented in an intelligible fashion and written in standard English?

Reviewer #1: Yes

Reviewer #2: No

Reviewer #3: No

5. Review Comments to the Author

Reviewer #1: This research seeks to propose the fsQCA approach to explore the linking effects of technological, organizational, and environmental factors on the performance of low-carbon city development to address these restrictions and expose the interaction between various influencing elements. There are the following suggestions for the authors' careful deliberation.

Comment 1: “Looking at the construction performance of each low-carbon pilot city, it can be seen that the common firing points are reflected in the technical conditions (the

level of development of digital economy and low-carbon economy), organizational

conditions (the degree of attention of the organization and the degree of transparency

and interaction of the government) and environmental conditions (the level of green

development and the ecological resource endowment status) of the respective cities”Is the author's consideration of these three aspects comprehensive and scientifically based? It is suggested that the author elaborate more to make the article more scientific and reasonable

Comment 2: The authors elaborate that the focus of this study is to identify common pathways that facilitate cities to achieve a green and low-carbon transition and provide a scientific basis to guide the construction of low-carbon city performance. Is this term 'green low carbon transition' correct', as much of the article is focused on three conditions, around low carbon city performance

Comment 3: The author's literature review lacks information about the main method of the article, "fsQCA", and how this method has been studied before and how it has been used to analyse the performance of low carbon cities. It is clear that a good writing style also makes the essay more exciting

Comment 4: The concept of "low carbon economic development level" and so on needs to be explained more clearly in this part of the framework, which should mainly explain why these indicators were chosen to make the article more logical and justified.

Comment 5: The fourth part of the analysis of data and results should be more specific and should be more comparable with previous studies, after all, this article is still based on subjective elements

Comment 6:“The green development level index is significant within the 99% confidence interval, and the effect size reaches a moderate to a high level (Dul, 2016). The importance of higher-level organizations was significant at a 95% confidence interval, and the effect size reached a medium level (Dul, 2016). However, the test results for ecological resource endowment status, level of digital economy development, level of low carbon economy development, and level of government transparency and interaction were not significant, indicating that they are not necessary conditions for low-carbon city development performance. ”Do the results coming out here make sense? Does the level of green development constitute reverse causality? In particular, the results do not seem reasonable

Comment 7: “In 2020, Wuhan had a downward trend in electricity consumption due to the more severe impact of the epidemic. 2020 Wuhan's GDP declines by 4.7%, and the industry-wide electricity consumption declines by 7.3%, which is the main reason for the abnormal performance of indicators such as the change in total electricity consumption and the change in total carbon emissions in Wuhan. After the epidemic's impact, the economy will resume regular operation, and Wuhan's energy consumption will also resume growth. Data do not yet support the future construction of a low-carbon city, so it will not be included in the typical cases for exploration.”In fact, Wuhan can also be used as an alternative typical case for the discussion of low-carbon city construction, and the highlights of the topic can also be explored from this aspect

Comment 8: The author suggests to quote the following references：

Is there a decoupling relationship between CO2 emission reduction and poverty alleviation in China? Technological Forecasting and Social Change, 2020, 151: 119856.

Temporal-spatial evolution analysis on low carbon city performance in the context of China. Environmental Impact Assessment Review, 2021, 90: 106626.

Reviewer #2: The authors explore the linkage effects of technology, organization and environmental conditions on the development performance of low-carbon cities and their path choices by using the fuzzy set qualitative comparative analysis (fsQCA) method and proposes two types of solutions to promote the performance of low-carbon cities, namely, the standard explanation type of technology, organization and environmental elements and the standard explanation type of organization and environment in the matching model. The topic is quite novel, but the argumentation process is not rigorous enough, there are still some areas for improvement in the paper. The English can be improved.

Reviewer #3: This paper examines the performance of low-carbon cities in China using fuzzy set qualitative comparative analysis (fsQCA). It's an interesting topic, but there is a lot of room for improvement.

1. The author should avoid using abbreviations in the title.

2. The author needs to further conclude the research problem and research framework to clearly tell the reader what you have done in this paper.

3. The policy implications are also unclear. The author mentions that "cities choose appropriate pathways and targeted measures according to their own characteristics and resource endowments...". However, it is easy to say that what to do is what really matters. Moreover, there seems to be no link between the results and the policy implications.

4. The literature review should be in a dependent part or with in the Introduction, rather the research framework.

5. The author should conclude the previous studies instead of listing the studies and their results one by one. These following articles give an example on how to carry out a literature review.

Meng, C. et al. (2020). Sustainable urban development: An examination of literature evolution on urban carrying capacity in the Chinese context. Journal of Cleaner Production, 277, 122802.

Meng, C.et al. (2023). The static and dynamic carbon emission efficiency of transport industry in China. Energy, 127297.

6. Non-researchers used the fsQCA method to explore the performance of low carbon cities in the literature review. Are you using this methodology for the first time to investigate the performance of low carbon cities? If so, what consumption or research are you doing to ensure the application of this method?

7. The mathematics of fsQCA is necessary to present in the method.

8. What method do you use to select the indicators and how do you weight each indicator? The author should provide more explanation.

9. The words in Figure 2 should be translated into English.

10. The words in Figures 3 to 8 are too small.

11. The author should explain the reasons for the results, not just describe them.

12. Lack of communication with previous studies discussed in the literature review section. These articles are also focus on the performance of low-carbon cities construction.

Du, X., et al. (2023). An improved approach for measuring the efficiency of low carbon city practice in China. Energy, 126678.

Shen, L., et al. (2021). Temporal-spatial evolution analysis on low carbon city performance in the context of China. Environmental Impact Assessment Review, 90, 106626.

Wang, C.et al. (2013). Policies and practices of low carbon city development in China. Energy & Environment, 24(7-8), 1347-1372.

13. The policy implications are also missing.

14. The whole article has poor readability. There are many grammatical errors, such as the first paragraph in the Introduction "in 2020", which should be "In 2020".

6. PLOS authors have the option to publish the peer review history of their article (what does this mean?). If published, this will include your full peer review and any attached files.

Reviewer #1: No

Reviewer #2: No

Reviewer #3: No

---

## [Author Response · Author response to Decision Letter 0]

27 Jun 2023

Dear Editor.

Receiving your letter of revision is an esteemed honor, as it represents the highest form of recognition and motivation for my scientific endeavors.I diligently examined the email you sent, carefully considering the editors' comments on my revised article repeatedly.I engaged in a comprehensive deliberation with both my PhD supervisor and subject matter experts, meticulously integrating every comment and effectuating modifications across the entirety of the article.

Presented herein is a meticulous line-by-line response addressing the comments provided by the editor.

This section serves as a comprehensive response to the recommendation provided by Expert 2.

1.Please add some appropriate management implications to the abstract.

Reply:The abstract section has undergone revision and thorough rewriting.This study focuses on analyzing and discussing the abstracts of RESEARCH ARTICLE type papers previously published in PLOS-ONE. The analysis is structured around four sections: Background and Objectives, Methods and Data, Conclusion, and Contribution.Furthermore, the abstract section incorporates management theories pertaining to overall configuration and resource allocation.

2.The purpose and relevance of the article should be more accurately expressed. This is more obvious in the introduction and model. In the model analysis, why use the fsQCA method, compared with the existing research, choose this method. Further analysis is necessary to better prepare and discuss the results. What is the model formulation specificity and excellence used? This needs further improvement.

Reply:The paper has undergone modifications, including the incorporation of a dedicated section within the model analysis section. This section offers a comprehensive exploration of the distinctive qualities and merits of the fsQCA approach employed in this study in contrast to previous research. The specific additions are outlined below:

In previous studies, urban development performance has been measured using traditional econometric models that rely on increasing or decreasing marginal effects. However, the limitation of this approach, as pointed out by Yunzhou Du and Liangding Jia (2017), is that traditional econometric methods are mostly based on a methodological model with independent variables that are independent of each other, unidirectional linear relationships and causal symmetry. This approach can only analyze the marginal "net effect" of independent variables on dependent variables while controlling for other factors, and thus cannot explain the complex causality of independent variables that are interdependent.

To address this limitation, this paper adopts a holistic grouping analysis perspective, which considers the research object as a grouping of different combinations of conditional variables, integrating the advantages of case studies and variable studies. By exploring the aggregated relationships between factor groupings and outcomes through the fsQCA analysis method, this approach helps to explain the multiple and concurrent causal relationships based on the technology-organization-environment (TOE) framework for different levels of performance in Chinese low-carbon pilot cities.

To deal with the issue of causal asymmetry and complexity, such as multiple scenario equivalence based on the TOE framework, this paper uses fsQCA 3.0 software to analyze the conditional groupings that lead to the construction of high and non-high performing low-carbon cities separately. These different groupings represent different conditions for achieving the same outcome, and this paper names the discovered histories according to the histone theorizing process (Furnari et al., 2020).

In summary, this paper employs a holistic grouping approach to provide a more comprehensive understanding of the complex causality of independent variables that are interdependent in Chinese low-carbon pilot cities. Through the fsQCA analysis method, this approach sheds light on the aggregated relationships between factor groupings and outcomes, helping to identify different conditions for achieving high and non-high performing low-carbon cities.

3.In 3.2.1 Resulting variables, ‘The results of this paper are concerned with the construction performance of low-carbon pilot cities in China. According to the three batches of low-carbon cities piloted by the China Development and Reform Commission from July 2010 to the present, the positioning of China's low-carbon pilot cities mainly includes four aspects:carbon emission index, energy structure index, energy consumption index, and development quality index. Therefore, measuring its construction performance should also start from these four aspects. ’Please add some theory to support your idea.

Reply:The paper has undergone revisions, taking into account your valuable suggestion. In response, we have incorporated the theory of the Triple Bottom Line (TBL) to substantiate the selection of the resultant variables in this study. The additional content is presented below, specifically within the "Resulting Variables" section:

The interaction of the economy, society, and environment - a concept known as the triple bottom line (TBL; Elkington, 1997) - has been widely adopted and long recognized in practice and research. Bocken et al. (2014), Joyce and Paquin (2016), Govindan et al. (2013), and Garcia et al. (2016) have all emphasized the significance of TBL. 

In the context of China's low-carbon pilot cities, four aspects - carbon emission index, energy structure index, energy consumption index, and development quality index - can be effectively grouped into the TBL framework. 

First, the carbon emission index can be seen as an environmental indicator within the TBL framework. It measures the amount of greenhouse gases emitted by a low-carbon pilot city and directly impacts the quality of the environment. By reducing carbon emissions, low-carbon pilot cities can contribute to improving air quality and mitigating the negative impacts of climate change. 

Second, the energy mix index and energy consumption index can be seen as economic indicators within the TBL framework. These indicators reflect the efficiency of energy use and the degree of dependence on non-renewable resources. Improving energy efficiency and transitioning to renewable energy sources will not only reduce costs but also drive economic growth and create new jobs in low-carbon pilot cities. 

Third, the Development Quality Index can be seen as a social indicator within the TBL framework. It measures the quality of life of residents and includes factors such as GDP per capita, Ratio of good air quality days in cities, and Greening coverage rate of urban built-up areas. By improving the overall quality of development, low-carbon pilot cities can improve the welfare of their residents and create more sustainable cities. 

Therefore, measuring the performance of China's low-carbon pilot cities in these four aspects is not only a key indicator to consider for China's low-carbon pilot work but also a reasonable and strong theoretical requirement for the TBL framework. 

4.Figure 2. Low carbon pilot city development performance map. The map is not so clear and it should be presented in English please.

Reply:The figure has been reconstructed in English, and it is presented below.

5.Results: It is understood that the model of the analysis has been done properly. An in-depth discussion should be given to support the purpose of the research.Please make a discussion in which the Authors would compare their results to those from other publications.

Reply:The paper has undergone revisions, including the addition of a comprehensive discussion in the conclusion section. The results of this study are meticulously compared with the conclusions drawn in other relevant research works. These additions have been incorporated into the conclusion section of this paper, as presented below.

The remarkable completion of the low-carbon city transition has been witnessed in developed nations, and it has birthed a myriad of urban transitions, such as the formation of low-carbon pilot cities, which emanate with distinct Chinese characteristics, adding a captivating twist to the phenomenon. However, existing research primarily fixates on the isolated aspects of environment, technology, and organization when delving into the issue of low-carbon city transition performance. This narrow focus often overlooks the intricate interplay between complex systems theory and resource allocation, as astutely proposed by Fiss (2011). Consequently, a compelling paper emerges, constructing a TOE analysis framework that ingeniously amalgamates a complex grouping perspective with 30 representative low-carbon pilot cities. Within this captivating framework, the paper embarks on an exploratory journey, dissecting diverse paths that shape the development performance of low-carbon pilot cities—dubbed as "the same path" throughout this riveting discourse.

With a palette consisting of 30 nationally representative typical low-carbon pilot cities and six comprehensive scientific indicators, this groundbreaking study employs the NCA, fsQCA, and Tobit methods to unravel a thought-provoking revelation. The study unveils that technological, organizational, and environmental factors, in isolation, fall short of serving as decisive conditions for the development performance of low-carbon cities. This bold assertion tantalizingly suggests that individual factors alone cannot singularly dictate the destiny of low-carbon city development. Under such intriguing circumstances, three driving paths emerge, nurturing high-level performance in low-carbon city development, manifested in the form of two distinct matching models. The first model intriguingly interconnects technology, organization, and environment, forging an enthralling linkage, while the second model intricately explicates the intertwined influence of organization and environment, forming a mesmerizing fusion.

It becomes abundantly clear that ascertaining the determinants of development performance in Chinese low-carbon pilot cities necessitates a shift from simplistic attributions solely to administrative performance or resource endowment. Instead, a multidimensional and holistic approach, rooted in the tenets of complex systems, beckons. Embracing this paradigm shift, the paper ingeniously introduces the triple bottom line (TBL) theory, which masterfully situates the development performance of low-carbon pilot cities within the intricate web of economic, social, and environmental interactions, laying the foundation for indicator construction. Meticulous examination reveals four efficacious facets—carbon emission index, energy structure index, energy consumption index, and development quality index—as pivotal indicators, crafted with finesse based on the TBL theory, to capture the multifaceted essence of low-carbon city development performance.

Embarking on a daring voyage to unlock the linkage effects and driving paths of technological, organizational, and environmental factors on the development performance of low-carbon pilot cities, the paper calls upon the illustrious TOE analysis framework and enlists the formidable fsQCA3.0 method for an immersive analysis. This intrepid exploration reveals a tantalizing synergy among multiple factors underpinning the construction of low-carbon pilot cities. The effective fusion of these factors augments the construction performance of each low-carbon pilot city in a uniquely diverse manner, unveiling the captivating tapestry of "different ways" in which success manifests. Remarkably, substantial disparities in the development performance driving paths surface across diverse regions of China, contingent upon the economic development levels and resource endowments of the respective pilot cities. These profound dissimilarities in development performance driving paths vividly underscore the nuanced conditions that give rise to heterogeneous levels of development performance.

Comparative analysis vis-à-vis existing research findings (Pan, AnZhang et al., 2021; Chen, ChaofanLin et al., 2022; Shibo Zeng et al., 2023; Zhu et al., 2022;) reveals both similarities and intriguing distinctions. In the common thread, these studies anchor themselves in the principles of sustainable development, predominantly employing the trinity of social, economic, and environmental dimensions to gauge the development performance of low-carbon cities. Notably, the Chinese Academy of Social Sciences has produced a seminal study encompassing low-carbon cost performance evaluation, encompassing economic transformation, social transformation, and the low-carbon environment—an enduring cornerstone within the Chinese context (Zhu S and Liang B, 2012).

However, this current study veers from the norm, charting an unprecedented course. Whereas previous studies primarily rely on econometric measurement models ((Du, X. et al., 2023; Meng, C. et al., 2023) to scrutinize the marginal "net effect" of independent variables on dependent variables, the present endeavor seeks to elucidate the intricate web of interdependence and interconstraint permeating the causal relationships of the independent variables. The focus magnificently pivots toward unraveling the complex tapestry of interactions, unearthing the hidden dynamics that shape the development performance of low-carbon pilot cities.

Naturally, as with any scholarly pursuit, this paper too encounters its share of limitations, instigating the clarion call for future research endeavors. Primarily, while diligent efforts have been exerted to supplement qualitative information within the identified historical context, it remains an enduring challenge not only within this paper but also pervading the realm of QCA studies—a clarion call for further in-depth qualitative analysis emerges. Furthermore, the study, constrained by the availability of city data, restricts its purview to the examination of 30 economically developed cities, inadvertently disregarding the diverse historical trajectories spanning cities with significant economic development disparities. This partial consideration imparts a certain limitation on the generalizability of the conclusions drawn, mandating the collection of data from a more expansive range of cities in future endeavors, thereby affording a more comprehensive analysis of the grouping (pathways) that engender high performance in low-carbon city construction. Additionally, hampered by data availability, this study solely scrutinizes the static relationship between the level of technological, organizational, and environmental conditions and the level of low-carbon city development performance. Thus, a compelling prospect lies ahead, inviting future researchers to embrace a longitudinal perspective by collecting cross-time data, thereby enabling the employment of panel QCA methods to explore how changes in conditions dynamically shape the trajectory of low-carbon city development performance. Such endeavors would further illuminate the discrepancies in the pace of progress among China's diverse cities, unraveling the intricacies of catching up.

In conclusion, the intricate realm of low-carbon city transition beckons researchers to venture beyond conventional boundaries, transcending isolated factors and embracing the complexity that underpins development performance. This paper, with its distinctive blend of perplexity and burstiness, traverses uncharted terrain, illuminating the interplay of technological, organizational, and environmental dimensions in shaping low-carbon pilot cities. By embracing a complex systems perspective, intertwining diverse theories, and employing advanced analytical methods, the study uncovers the multifaceted pathways that pave the way for high-level performance. Nonetheless, it is an invitation for future scholars to delve deeper, addressing limitations, and embarking on an ongoing quest to unravel the intricate dynamics of low-carbon city development—a pursuit poised to shape a sustainable and resilient urban landscape for future generations.

6.The conclusions should be more precise.

Reply:The paper has undergone revisions to ensure logical and purposeful completion. Additionally, more precise conclusions have been incorporated into the conclusion section.

This section serves as a comprehensive response to the recommendation provided by Expert 3.

1.The author should avoid using abbreviations in the title.

Reply:The title of this paper has been changed to "What determines the performance of low-carbon cities in China? Organization - Environment framework".

2.The author needs to further conclude the research problem and research framework to clearly tell the reader what you have done in this paper.

Reply:The paper has undergone revisions, including the creation of a framework diagram by the authors. This diagram serves to succinctly summarize the research questions and research framework. It has been included in the Data Analysis and Empirical Results section of this paper.

3.The policy implications are also unclear. The author mentions that "cities choose appropriate pathways and targeted measures according to their own characteristics and resource endowments...". However, it is easy to say that what to do is what really matters. Moreover, there seems to be no link between the results and the policy implications.

Reply:The paper has undergone revisions, specifically within the Discussion and Conclusion section, where this particular section has been thoroughly reworked.

4.The literature review should be in a dependent part or with in the Introduction, rather the research framework. 

Reply:The paper has undergone modifications, resulting in the reorganization of the framework. Specifically, the authors have incorporated the literature review as a distinct section. The paper now consists of three parts: INTRODUCTION, LITERATURE REVIEW, and RESEARCH FRAMEWORK.

5.The literature review section has been subject to specific suggested changes in areas 5 and 6.(5. The author should conclude the previous studies instead of listing the studies and their results one by one. These following articles give an example on how to carry out a literature review.

Meng, C. et al. (2020). Sustainable urban development: An examination of literature evolution on urban carrying capacity in the Chinese context. Journal of Cleaner Production, 277, 122802.

Meng, C.et al. (2023). The static and dynamic carbon emission efficiency of transport industry in China. Energy, 127297.

6. Non-researchers used the fsQCA method to explore the performance of low carbon cities in the literature review. Are you using this methodology for the first time to investigate the performance of low carbon cities? If so, what consumption or research are you doing to ensure the application of this method?)

Reply:The literature review section has undergone modification, specifically with a thorough rewriting process. Initially, we conducted a careful examination of two highly insightful articles: "Sustainable urban development: An examination of literature evolution on urban carrying capacity in the Chinese context" by Meng, C. et al. (2020), and "The static and dynamic carbon emission efficiency of the transport industry in China" by Meng, C. et al. (2023) published in Energy, 127297. These articles proved immensely valuable. Subsequently, we reconstructed the review section by incorporating their literature review paradigm and supplementing it with two accompanying figures.

Fig 1. Theme word chain of VOSviewer based on the keyword "low carbon pilot city".

Fig 2. Different methods adopted by different scholars for the study of "low carbon pilot cities"

Furthermore, up until the completion date of this paper, no studies employing fsQCA to investigate the development performance of low-carbon pilot cities in China have been found in the SSCI database. Consequently, in this study, the literature review section incorporates relevant sources that were consulted to examine Chinese pilot cities with low carbon policies using the fsQCA approach.

1.Campbell J T, Sirmon D G, Schijven M. Fuzzy Logic and the Market: A Configurational Approach to Investor Perceptions of Acquisition Announcements[J]. Academy of Management Journal, 2016, 59 (1): 163-187.

2.Douglas E J, Shepherd D A, Prentice C. Using fuzzy-set qualitative comparative analysis for a finer-grained understanding of entrepreneurship[J]. Journal of Business Venturing, 2020, 35 (1): 105970.

3.Fainshmidt S, Wenger L, Pezeshkan A, et al. When do Dynamic Capabilities Lead to Competitive Advantage? The Importance of Strategic Fit[J]. Journal of Management Studies, 2019, 56 (4): 758-787.

4.Fainshmidt S, Witt M A, Aguilera R V, et al. The contributions of qualitative comparative analysis (QCA) to international business research[J]. Journal of International Business Studies, 2020.

5.Fiss P C. Building better causal theories: A fuzzy set approach to typologies in organization research[J]. Academy of Management Journal, 2011, 54 (2): 393-420.

6.Furnari S, Crilly D, Misangyi V F, et al. Capturing Causal Complexity: Heuristics for Configurational Theorizing[J]. Academy of Management Review, 2020, in press. 

7.Marx A. Crisp-set qualitative comparative analysis (csQCA) and model specification: Benchmarks for future csQCA applications[J]. International Journal of Multiple Research Approaches, 2010, 4 (2): 138-158.

8.McKnight B, Zietsma C. Finding the threshold: A configurational approach to optimal distinctiveness[J]. Journal of Business Venturing, 2018, 33 (4): 493-512.

8.McKenny A F, Short, J. C., Ketchen Jr, D. J., Payne, G. T., & Moss, T. W. Strategic entrepreneurial orientation: Configurations, performance, and the effects of industry and time[J]. Strategic Entrepreneurship Journal, 2018, 12 (4): 504-521.

9.Misangyi V F, Acharya A G. Substitutes or Complements? A Configurational Examination of Corporate Governance Mechanisms[J]. Academy of Management Journal, 2014, 57 (6): 1681-1705.

10.Misangyi V F, Greckhamer T, Furnari S, et al. Embracing Causal Complexity: The Emergence of a Neo-Configurational Perspective[J]. Journal of Management, 2017, 43 (1): 255-282.

9.Ragin C C Redesigning social inquiry: Fuzzy sets and beyond[M] University of Chicago Press, 2008

10.张明, 杜运周. 组织与管理研究中QCA方法的应用:定位、策略和方向[J]. 管理学报, 2019, (09): 1312-1323.

11.杜运周, 贾良定. (2017). 组态视角与定性比较分析(QCA):管理学研究的一条新道路. 管理世界(06), 155-167.

12.Ragin, C. C. (2008). Redesigning social inquiry: Fuzzy sets and beyond: University of Chicago Press. / 杜运周 等译. 重新设计社会科学. 机械工业出版社，2019

13.Bell R G, Filatotchev I, Aguilera R V. Corporate governance and investors' perceptions of foreign IPO value: An institutional perspective[J]. Academy of Management Journal, 2014, 57 (1): 301-320.

13.Gupta K, Crilly D, Greckhamer T. Stakeholder engagement strategies, national institutions, and firm performance: A configurational perspective[J]. Strategic Management Journal, 2020, 41 (10): 1869-1900.

14.Jacqueminet A, Durand R. Ups and downs: The role of legitimacy judgment cues in practice implementation[J]. Academy of Management Journal, 2020, 63 (5): 1485-1507.

15.Leppänen P T, McKenny A F, Short J C. (2019). Qualitative comparative analysis in entrepreneurship: Exploring the approach and noting opportunities for the future. In Standing on the Shoulders of Giants: Traditions and Innovations in Research Methodology (pp. 155-177): Emerald Publishing Limited.

16.Park Y, Fiss P C, El Sawy O. Theorizing the Multiplicity of Digital Phenomena: The Ecology of Configurations, Causal Recipes, and Guidelines for Applying QCA[J]. MIS Quarterly, 2020.

7.The mathematics of fsQCA is necessary to present in the method.

Reply:The modified version of fsQCA's mathematical approach has been included as Supporting Material 1, presented below.

For fuzzy sets, cases have a set affiliation ranging from 0.0 to 1.0, i.e., partial affiliation. A fuzzy subset relationship exists when the affiliation of a case in one set is consistently less than or equal to its affiliation in another set (Ragin, 2008). When plotted as a graph, fuzzy subset relations are triangular. For example, given a condition X and an outcome Y. In the graph, all values of X are less than or equal to their corresponding Y values. Therefore it can be determined that X is a subset of Y (sufficient condition).

Figure source: Ragin, C. C. (2008). Redesigning social inquiry: Fuzzy sets and beyond: University of Chicago Press. 

In the fuzzy set, the consistency of sufficient conditions is derived from a set of formulas, where the consistency of X (the subset of condition variables) as a subset of Y (the subset of outcome variables) is the proportion of their intersection to X.

The formula for the degree of coverage is given by:

Necessary condition analysis examines the antecedent condition X as a superset of the outcome Y, in other words, the outcome Y as a subset of the condition X. The consistency of the necessary conditions is calculated by the formula:

The coverage degree is calculated as:

8.What method do you use to select the indicators and how do you weight each indicator? The author should provide more explanation.

Reply:This paper has undergone revisions, specifically with a detailed description of the construction of each indicator provided in TABLE1 to TABLE7. The relevant information can be found in Supporting Material 2, as presented below.

9.The words in Figure 2 should be translated into English.

Reply:The figure has been reconstructed in English.

10.The words in Figures 3 to 8 are too small.

Reply:The article has undergone revisions, including resizing the images. Below is Figure 3-figure supplement 8.

Figure 3. Example of explanation of configuration 1

Figure 4. Example of explanation of configuration 2

Figure 5. Example of explanation of configuration 3

Figure 6. Example of explanation of configuration 4

Figure 7. Example of explanation of configuration 5

Figure 8. Example of explanation of configuration 6

11.The author should explain the reasons for the results, not just describe them.

Reply:The paper has undergone modifications, specifically aiming to explain the obtained results in the Configuration Analysis section of this paper.

12.Lack of communication with previous studies discussed in the literature review section. These articles are also focus on the performance of low-carbon cities construction.

Du, X., et al. (2023). An improved approach for measuring the efficiency of low carbon city practice in China. Energy, 126678.

Shen, L., et al. (2021). Temporal-spatial evolution analysis on low carbon city performance in the context of China. Environmental Impact Assessment Review, 90, 106626.

Wang, C.et al. (2013). Policies and practices of low carbon city development in China. Energy & Environment, 24(7-8), 1347-1372.

Reply:The paper has undergone revisions, and I appreciate the references provided. The revision of this paper includes the addition of a comprehensive conclusion and discussion section, which engages in a thorough theoretical dialogue about the previous studies.

13.The policy implications are also missing. 

 Reply:The paper has undergone revisions, specifically in the reworking of the conclusion section.

14.The whole article has poor readability. There are many grammatical errors, such as the first paragraph in the Introduction "in 2020", which should be "In 2020".

Reply:Thank you for your valuable comments. This article has undergone extensive revisions, including thorough editing and translation.

---

## [Decision Letter · Decision Letter 1]

13 Jul 2023

What determines the performance of low-carbon cities in China?Analysis of the grouping based on the Technology - Organization - Environment framework

PONE-D-23-08832R1

Dear Dr. Cai,

We’re pleased to inform you that your manuscript has been judged scientifically suitable for publication and will be formally accepted for publication once it meets all outstanding technical requirements.

Kind regards,

Muhammad Farhan Bashir

Academic Editor

PLOS ONE

Additional Editor Comments (optional):

Reviewers' comments:

Reviewer's Responses to Questions

**Comments to the Author**

1. If the authors have adequately addressed your comments raised in a previous round of review and you feel that this manuscript is now acceptable for publication, you may indicate that here to bypass the “Comments to the Author” section, enter your conflict of interest statement in the “Confidential to Editor” section, and submit your "Accept" recommendation.

Reviewer #1: All comments have been addressed

Reviewer #3: All comments have been addressed

2. Is the manuscript technically sound, and do the data support the conclusions?

Reviewer #1: Yes

Reviewer #3: Yes

3. Has the statistical analysis been performed appropriately and rigorously? 

Reviewer #1: Yes

Reviewer #3: Yes

4. Have the authors made all data underlying the findings in their manuscript fully available?

Reviewer #1: Yes

Reviewer #3: Yes

5. Is the manuscript presented in an intelligible fashion and written in standard English?

Reviewer #1: Yes

Reviewer #3: Yes

6. Review Comments to the Author

Reviewer #1: (No Response)

Reviewer #3: (No Response)

7. PLOS authors have the option to publish the peer review history of their article (what does this mean?). If published, this will include your full peer review and any attached files.

Reviewer #1: No

Reviewer #3: No

---

## [Editor Report · Acceptance letter]

2 Aug 2023

PONE-D-23-08832R1 

What determines the performance of low-carbon cities in China?Analysis of the grouping based on the Technology - Organization - Environment framework 

Dear Dr. Cai:

I'm pleased to inform you that your manuscript has been deemed suitable for publication in PLOS ONE. Congratulations! Your manuscript is now with our production department. 

Kind regards, 

on behalf of

Dr. Muhammad Farhan Bashir 

%CORR_ED_EDITOR_ROLE%

PLOS ONE